# The Hessian Screening Rule

**Johan Larsson**
Department of Statistics
Lund University
johan.larsson@stat.lu.se

**Jonas Wallin**
Department of Statistics
Lund University
jonas.wallin@stat.lu.se

## Abstract

Predictor screening rules, which discard predictors before fitting a model, have had considerable impact on the speed with which sparse regression problems, such as the lasso, can be solved. In this paper we present a new screening rule for solving the lasso path: the Hessian Screening Rule. The rule uses second-order information from the model to provide both effective screening, particularly in the case of high correlation, as well as accurate warm starts. The proposed rule outperforms all alternatives we study on simulated data sets with both low and high correlation for $\ell_1$-regularized least-squares (the lasso) and logistic regression. It also performs best in general on the real data sets that we examine.

## 1 Introduction

High-dimensional data, where the number of features ($p$) exceeds the number of observations ($n$), poses a challenge for many classical statistical models. A common remedy for this issue is to regularize the model by penalizing the regression coefficients such that the solution becomes sparse. A popular choice of such a penalization is the $\ell_1$-norm, which, when the objective is least-squares, leads to the well-known lasso [1]. More specifically, we will focus on the following convex optimization problem:

$$\underset{\beta \in \mathbb{R}^p}{\text{minimize}} \left\{ f(\beta; X) + \lambda \|\beta\|_1 \right\}, \tag{1}$$

where $f(\beta; X)$ is smooth and convex. We let $\hat{\beta}$ be the solution vector for this problem and, abusing notation, equivalently let $\hat{\beta} : \mathbb{R} \mapsto \mathbb{R}^p$ be a function that returns this vector for a given $\lambda$. Our focus lies in solving (1) along a regularization path $\lambda_1, \lambda_2 \ldots, \lambda_m$ with $\lambda_1 \geq \lambda_2 \geq \cdots \geq \lambda_m$. We start the path at $\lambda_{\max}$, which corresponds to the null (all-sparse) model[1], and finish at some fraction of $\lambda_{\max}$ for which the model is either almost saturated (in the $p \geq n$ setting), or for which the solution approaches the ordinary least-squares estimate. The motivation for this focus is that the optimal $\lambda$ is typically unknown and must be estimated through model tuning, such as cross-validation. This involves repeated refitting of the model to new batches of data, which is computationally demanding.

Fortunately, the introduction of so-called *screening rules* has improved this situation remarkably. Screening rules use tests that screen and possibly discard predictors from the model *before* it is fit, which effectively reduces the dimensions of the problem and leads to improvements in performance and memory usage. There are, generally speaking, two types of screening rules: *safe* and *heuristic* rules. Safe rules guarantee that discarded predictors are inactive at the optimum—heuristic rules do not and may therefore cause violations: discarding active predictors. The possibility of violations mean that heuristic methods need to validate the solution through checks of the Karush–Kuhn–Tucker (KKT) optimality conditions after optimization has concluded and, whenever there are violations, re-run optimization, which can be costly particularly because the KKT checks themselves are expensive. This means that the distinction between safe and heuristic rules only matters in regards to algorithmic

---

[1] $\lambda_{\max}$ is in fact available in closed form—for the lasso it is $\max_j |x_j^T y|$.

36th Conference on Neural Information Processing Systems (NeurIPS 2022).

details—all heuristic methods that we study here use KKT checks to catch these violations, which means that these methods are in fact also safe.

Screening rules can moreover also be classified as *basic*, *sequential*, or *dynamic*. Basic rules screen predictors based only on information available from the null model. Sequential rules use information from the previous step(s) on the regularization path to screen predictors for the next step. Finally, dynamic rules screen predictors during optimization, reducing the set of screened predictors repeatedly throughout optimization.

Notable examples of safe rules include the basic SAFE rule [2], the sphere tests [3], the R-region test [4], Slores [5], Gap Safe [6, 7], and Dynamic Sasvi [8]. There is also a group of dual polytope projection rules, most prominently Enhanced Dual Polytope Projection (EDPP) [9]. As noted by Fercoq, Gramfort, and Salmon [6], however, the sequential version of EDPP relies on exact knowledge of the optimal solution at the previous step along the path to be safe in practice, which is only available for $\lambda_{\max}$. Among the heuristic rules, we have the Strong Rule [10], SIS [11], and ExSIS [12]. But the latter two of these are not sequential rules and solve a potentially reduced form of the problem in (1)—we will not discuss them further here. In addition to these two types of rules, there has also recently been attempts to combine safe and heuristic rules into so-called hybrid rules [13].

There are various methods for employing these rules in practice. Of particular interest are so-called *working set* strategies, which use a subset of the screened set during optimization, iteratively updating the set based on some criterion. Tibshirani et al. [10] introduced the first working set strategy, which we in this paper will refer to simply as the *working set strategy*. It uses the set of predictors that have ever been active as an initial working set. After convergence on this set, it checks the KKT optimality conditions on the set of predictors selected by the strong rule, and then adds predictors that violate the conditions to the working set. This procedure is then repeated until there are no violations, at which point the optimality conditions are checked for the entire set, possibly triggering additional iterations of the procedure. Blitz [14] and Celer [15] are two other methods that use both Gap Safe screening and working sets. Instead of choosing previously active predictors as a working set, however, both Blitz and Celer assign priorities to each feature based on how close each feature is of violating the Gap Safe check and construct the working set based on this prioritization. In addition to this, Celer uses dual point acceleration to improve Gap Safe screening and speed up convergence. Both Blitz and Celer are heuristic methods.

One problem with current screening rules is that they often become conservative—including large numbers of predictors into the screened set—when dealing with predictors that are strongly correlated. Tibshirani et al. [10], for instance, demonstrated this to be the case with the strong rule, which was the motivation behind the working set strategy. (See Appendix F.4 for additional experiments verifying this). Yet because the computational complexity of the KKT checks in the working set strategy still depends on the strong rule, the effectiveness of the rule may nevertheless be hampered in this situation. A possible and—as we will soon show—powerful solution to this problem is to make use of the second-order information available from (1), and in this paper we present a novel screening rule based on this idea. Methods using second-order information (the Hessian) are often computationally infeasible for high-dimensional problems. We utilize two properties of the problem to remedy this issue: first, we need only to compute the Hessian for the active set, which is often much smaller than the full set of predictors. Second, we avoid constructing the Hessian (and it's inverse) from scratch for each $\lambda$ along the path, instead updating it sequentially by means of the Schur complement. The availability of the Hessian also enables us to improve the warm starts (the initial coefficient estimate at the start of each optimization run) used when fitting the regularization path, which plays a key role in our method.

We present our main results in Section 3, beginning with a reformulation of the strong rule and working set strategy before we arrive at the screening rule that represents the main result of this paper. In Section 4, we present numerical experiments on simulated and real data to showcase the effectiveness of the screening rule, demonstrating that the rule is effective both when $p \gg n$ and $n \gg p$, out-performing the other alternatives that we study. Finally, in Section 5 we wrap up with a discussion on these results, indicating possible ways in which they may be extended.

## 2 Preliminaries

We use lower-case letters to denote scalars and vectors and upper-case letters for matrices. We use $\mathbf{0}$ and $\mathbf{1}$ to denote vectors with elements all equal to 0 or 1 respectively, with dimensions inferred from context. Furthermore, we let sign be the standard signum function with domain $\{-1, 0, 1\}$, allowing it to be overloaded for vectors.

Let $c(\lambda) := -\nabla_\beta f\big(\hat{\beta}(\lambda); X\big)$ be the negative gradient, or so-called *correlation*, and denote $\mathcal{A}_\lambda = \{i : |c(\lambda)_i| > \lambda\}$ as the *active set* at $\lambda$: the support set of the non-zero regression coefficients corresponding to $\hat{\beta}(\lambda)$. In the interest of brevity, we will let $\mathcal{A} := \mathcal{A}_\lambda$. We will consider $\beta$ a solution to (1) if it satisfies the stationary criterion

$$\mathbf{0} \in \nabla_\beta f(\beta; X) + \lambda \partial. \tag{2}$$

Here $\partial$ is the subdifferential of $\|\beta\|_1$, defined as

$$\partial_j \in \begin{cases} \{\operatorname{sign}(\hat{\beta}_j)\} & \text{if } \hat{\beta}_j \neq 0, \\ [-1, 1] & \text{otherwise.} \end{cases}$$

This means that there must be a $\tilde{\partial} \in \partial$ for a given $\lambda$ such that

$$\nabla_\beta f(\beta; X) + \lambda \tilde{\partial} = \mathbf{0}. \tag{3}$$

## 3 Main Results

In this section we derive the main result of this paper: the Hessian screening rule. First, however, we now introduce a non-standard perspective on screening rules. In this approach, we note that (2) suggests a simple and general formulation for a screening rule, namely: we substitute the gradient vector in the optimality condition of a $\ell_1$-regularized problem with an estimate. More precisely, we discard the $j$th predictor for the problem at a given $\lambda$ if the magnitude of the $j$th component of the gradient vector estimate is smaller than this $\lambda$, that is

$$|\tilde{c}(\lambda)_j| < \lambda. \tag{4}$$

In the following sections, we review the strong rule and working set method for this problem from this perspective, that is, by viewing both methods as gradient approximations. We start with the case of the standard lasso ($\ell_1$-regularized least-squares), where we have $f(\beta; X) = \frac{1}{2}\|X\beta - y\|_2^2$.

### 3.1 The Strong Rule

The sequential strong rule for $\ell_1$-penalized least-squares regression [10] discards the $j$th predictor at $\lambda = \lambda_{k+1}$ if

$$\left| x_j^T (X\hat{\beta}(\lambda_k) - y) \right| = |c(\lambda_k)_j| < 2\lambda_{k+1} - \lambda_k.$$

This is equivalent to checking that

$$\tilde{c}^S(\lambda_{k+1}) = c(\lambda_k) + (\lambda_k - \lambda_{k+1})\operatorname{sign}(c(\lambda_k)) \tag{5}$$

satisfies (4). The strong rule gradient approximation (5) is also known as the *unit bound*, since it assumes the gradient of the correlation vector to be bounded by one.

### 3.2 The Working Set Method

A simple but remarkably effective alternative to direct use of the strong rule is the working set heuristic [10]. It begins by estimating $\beta$ at the $(k + 1)$th step using only the coefficients that have been previously active at any point along the path, i.e. $\mathcal{A}_{1:k} = \cup_{i=1}^k \mathcal{A}_i$. The working set method can be viewed as a gradient estimate in the sense that

$$\tilde{c}^W(\lambda_{k+1}) = X^T \left( y - X_{\mathcal{A}_{1:k}} \tilde{\beta}(\lambda_{k+1}, \mathcal{A}_{1:k}) \right) = -\nabla f\big(\tilde{\beta}(\lambda_{k+1}, \mathcal{A}_{1:k}); X\big),$$

where $\tilde{\beta}(\lambda, \mathcal{A}) = \arg\min_\beta \frac{1}{2}\|y - X_\mathcal{A}\beta\|^2 + \lambda|\beta|$.

### 3.3 The Hessian Screening Rule

We have shown that both the strong screening rule and the working set strategy can be expressed as estimates of the correlation (negative gradient) for the next step of the regularization path. As we have discussed previously, however, basing this estimate on the strong rule can lead to conservative approximations. Fortunately, it turns out that we can produce a better estimate by utilizing second-order information.

We start by noting that (3), in the case of the standard lasso, can be formulated as

$$\begin{bmatrix} X_{\mathcal{A}}^T X_{\mathcal{A}} & X_{\mathcal{A}}^T X_{\mathcal{A}^c} \\ X_{\mathcal{A}^c}^T X_{\mathcal{A}} & X_{\mathcal{A}^c}^T X_{\mathcal{A}^c} \end{bmatrix} \begin{bmatrix} \hat{\beta}_{\mathcal{A}} \\ 0 \end{bmatrix} + \lambda \begin{bmatrix} \text{sign}(\hat{\beta}(\lambda)_{\mathcal{A}}) \\ \partial_{\mathcal{A}^c} \end{bmatrix} = \begin{bmatrix} X_{\mathcal{A}}^T y \\ X_{\mathcal{A}^c}^T y \end{bmatrix},$$

and consequently that

$$\hat{\beta}(\lambda)_{\mathcal{A}} = (X_{\mathcal{A}}^T X_{\mathcal{A}})^{-1} \big( X_{\mathcal{A}}^T y - \lambda \, \text{sign}\,(\hat{\beta}_{\mathcal{A}}) \big).$$

Note that, for an interval $[\lambda_l, \lambda_u]$ in which the active set is unchanged, that is, $\mathcal{A}_\lambda = \mathcal{A}$ for all $\lambda \in [\lambda_u, \lambda_k]$, then $\hat{\beta}(\lambda)$ is a continuous linear function in $\lambda$ (Theorem 3.1)[2].

**Theorem 3.1.** *Let $\hat{\beta}(\lambda)$ be the solution of* (1) *where $f(\beta; X) = \frac{1}{2}\|X\beta - y\|_2^2$. Define*

$$\hat{\beta}^{\lambda^*}(\lambda)_{A_{\lambda^*}} = \hat{\beta}(\lambda^*)_{\mathcal{A}_{\lambda^*}} - (\lambda^* - \lambda)\, (X_{\mathcal{A}_{\lambda^*}}^T X_{\mathcal{A}_{\lambda^*}})^{-1} \text{sign}\,(\hat{\beta}(\lambda^*)_{\mathcal{A}_{\lambda^*}})$$

*and $\hat{\beta}^{\lambda^*}(\lambda)_{\mathcal{A}_{\lambda^*}^c} = 0$. If it for $\lambda \in [\lambda_0, \lambda^*]$ holds that (i) $\text{sign}\,(\hat{\beta}^{\lambda^*}(\lambda)) = \text{sign}\,(\hat{\beta}(\lambda^*))$ and (ii) $\max |\nabla f(\hat{\beta}^{\lambda^*}(\lambda))_{\mathcal{A}_{\lambda^*}}| < \lambda$, then $\hat{\beta}(\lambda) = \hat{\beta}^{\lambda^*}(\lambda)$ for $\lambda \in [\lambda_0, \lambda^*]$.*

See Appendix A for a full proof. Using Theorem 3.1, we have the following second-order approximation of $c(\lambda_{k+1})$:

$$\hat{c}^H(\lambda_{k+1}) = -\nabla f\big(\hat{\beta}^{\lambda_k}(\lambda_{k+1})_{A_{\lambda_k}}\big) = c(\lambda_k) + (\lambda_{k+1} - \lambda_k) X^T X_{\mathcal{A}_k} (X_{\mathcal{A}_k}^T X_{\mathcal{A}_k})^{-1} \text{sign}\,(\hat{\beta}(\lambda_k)_{\mathcal{A}_k}). \tag{6}$$

*Remark* 3.2. If no changes in the active set occur in $[\lambda_{k+1}, \lambda_k]$, (6) is in fact an exact expression for the correlation at the next step, that is, $\hat{c}^H(\lambda_{k+1}) = c(\lambda_{k+1})$.

One problem with using the gradient estimate in (6) is that it is expensive to compute due to the inner products involving the full design matrix. To deal with this, we use the following modification, in which we restrict the computation of these inner products to the set indexed by the strong rule, assuming that predictors outside this set remain inactive:

$$\tilde{c}^H(\lambda_{k+1})_j := \begin{cases} \lambda_{k+1} \text{sign}\,\hat{\beta}(\lambda_k)_j & \text{if } j \in \mathcal{A}_{\lambda_k}, \\ 0 & \text{if } |\tilde{c}^S(\lambda_{k+1})_j| < \lambda_{k+1} \text{ and } j \notin \mathcal{A}_{\lambda_k}, \\ \hat{c}^H(\lambda_{k+1})_j & \text{else.} \end{cases}$$

For high-dimensional problems, this modification leads to large computational gains and seldom proves inaccurate, given that the strong rule only rarely causes violations [10]. Lastly, we make one more adjustment to the rule, which is to add a proportion of the unit bound (used in the strong rule) to the gradient estimate:

$$\check{c}^H(\lambda_{k+1})_j := \tilde{c}^H(\lambda_{k+1})_j + \gamma(\lambda_{k+1} - \lambda_k)\,\text{sign}(c(\lambda_k)_j),$$

where $\gamma \in \mathbb{R}_+$. Without this adjustment there would be no upwards bias on the estimate, which would cause more violations than would be desirable. In our experiments, we have used $\gamma = 0.01$, which has worked well for most problems we have encountered. This finally leads us to the *Hessian screening rule*: discard the $j$th predictor at $\lambda_{k+1}$ if $|\check{c}^H(\lambda_{k+1})_j| < \lambda_{k+1}$.

We make one more modification in our implementation of the Hessian Screening Rule, which is to use the union of the ever-active predictors and those screened by the screening rule as our final set of screened predictors. We note that this is a marginal improvement to the rule, since violations of the rule are already quite infrequent. But it is included nonetheless, given that it comes at no cost and occasionally prevents violations.

---

[2]This result is not a new discovery [16], but is included here for convenience because the following results depend on it.

As an example of how the Hessian Screening Rule performs, we examine the screening performance of several different strategies. We fit a full regularization path to a design with $n = 200$, $p = 20\,000$, and pairwise correlation between predictors of $\rho$. (See Section 4 and Appendix F.4 for more information on the setup.) We compute the average number of screened predictors across iterations of the coordinate descent solver. The results are displayed in Figure 1 and demonstrate that our method gracefully handles high correlation among predictors, offering a screened set that is many times smaller than those produced by the other screening strategies. In Appendix F.4 we extend these results to $\ell_1$-regularized logistic regression as well and report the frequency of violations.

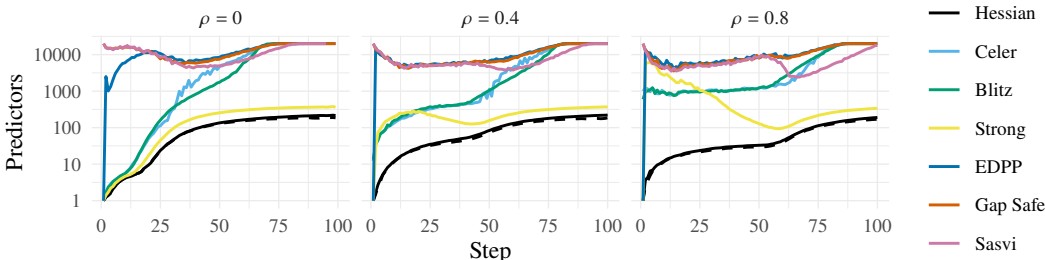

Figure 1: The number of predictors screened (included) for when fitting a regularization path of $\ell_1$-regularized least-squares to a design with varying correlation ($\rho$), $n = 200$, and $p = 20000$. The values are averaged over 20 repetitions. The minimum number of active predictors at each step across iterations is given as a dashed line. Note that the y-axis is on a $\log_{10}$ scale.

Recall that the Strong rule bounds its gradient of the correlation vector estimate at one. For the Hessian rule, there is no such bound. This means that it is theoretically possible for the Hessian rule to include more predictors than the Strong rule[3]. In fact, it is even possible to design special cases where the Hessian rule could be more conservative than the Strong rule. In practice, however, we have not encountered any situation in which this is the case.

### 3.3.1 Updating the Hessian

A potential drawback to using the Hessian screening rule is the computational costs of computing the Hessian and its inverse. Let $\mathcal{A}_k$ be the active set at step $k$ on the lasso path. In order to use the Hessian screening rule we need $H_k^{-1} = (X_{\mathcal{A}_k}^T X_{\mathcal{A}_k})^{-1}$. Computing $(X_{\mathcal{A}_k}^T X_{\mathcal{A}_k})^{-1}$ directly, however, has numerical complexity $O(|\mathcal{A}_k|^3 + |\mathcal{A}_k|^2 n)$. But if we have stored $(H_{k-1}^{-1}, H_{k-1})$ previously, we can utilize it to compute $(H_k^{-1}, H_k)$ more efficiently via the so-called sweep operator [17]. We outline this technique in Algorithm 1 (Appendix B). The algorithm has a reduction step and an augmentation step; in the reduction step, we reduce the Hessian and its inverse to remove the presence of any predictors that are no longer active. In the augmentation step, we update the Hessian and its inverse to account for predictors that have just become active.

The complexity of the steps depends on the size of the sets $\mathcal{C} = \mathcal{A}_{k-1} \setminus \mathcal{A}_k, \mathcal{D} = \mathcal{A}_k \setminus \mathcal{A}_{k-1}$, and $\mathcal{E} = \mathcal{A}_k \cap \mathcal{A}_{k-1}$ The complexity of the reduction step is $O(|\mathcal{C}|^3 + |\mathcal{C}|^2|\mathcal{E}| + |\mathcal{C}||\mathcal{E}|^2)$ and the complexity of the augmentation step is $O(|\mathcal{D}|^2 n + n|\mathcal{D}||\mathcal{E}| + |\mathcal{D}|^2|\mathcal{E}| + |\mathcal{D}|^3)$ since $n \geq \max(|\mathcal{E}|, |\mathcal{D}|)$. An iteration of Algorithm 1 therefore has complexity $O(|\mathcal{D}|^2 n + n|\mathcal{D}||\mathcal{E}| + |\mathcal{C}|^3 + |\mathcal{C}||\mathcal{E}|^2)$.

In most applications, the computationally dominant term will be $n|\mathcal{D}||\mathcal{E}|$ (since, typically, $n > |\mathcal{E}| > \mathcal{D} > \mathcal{C}$) which could be compared to evaluating the gradient for $\beta_{\mathcal{A}_k}$, which is $n(|\mathcal{D}| + |\mathcal{E}|)$ when $\beta_{\mathcal{A}_k^c} = 0$. Note that we have so far assumed that the inverse of the Hessian exists, but this need not be the case. To deal with this issue we precondition the Hessian. See Appendix C for details.

### 3.3.2 Warm Starts

The availability of the Hessian and its inverse offers a coefficient warm start that is more accurate than the standard, naive, approach of using the estimate from the previous step. With the Hessian screening rule, we use the following warm start.

$$\hat{\beta}(\lambda_{k+1})_{\mathcal{A}_k} := \hat{\beta}(\lambda_k)_{\mathcal{A}_k} + (\lambda_k - \lambda_{k+1})H_{\mathcal{A}_k}^{-1} \operatorname{sign}\left(\hat{\beta}(\lambda_k)_{\mathcal{A}_k}\right), \tag{7}$$

---

[3]The chance of this happening is tied to the setting of $\gamma$.

where $H_{\mathcal{A}_k}^{-1}$ is the Hessian matrix for the differentiable part of the objective. Our warm start is equivalent to the one used in Park and Hastie [18], but is here made much more efficient due due to the efficient updates of the Hessian and its inverse that we use.

*Remark* 3.3. The warm start given by (7) is the exact solution at $\lambda_k$ if the active set remains constant in $[\lambda_{k+1}, \lambda_k]$.

As a first demonstration of the value of this warm start, we look at two data sets: *YearPredicitionMSD* and *colon-cancer*. We fit a full regularization path using the setup as outlined in Section 4, with or without Hessian warm starts. For YearPredictionMSD we use the standard lasso, and for colon-cancer $\ell_1$-regularized logistic regression.

The Hessian warm starts offer sizable reductions in the number of passes of the solver (Figure 2), for many steps requiring only a single pass to reach convergence. On inspection, this is not a surprising find. There are no changes in the active set for many of these steps, which means that the warm start is almost exact—"almost" due to the use of a preconditioner for the Hessian (see Appendix C).

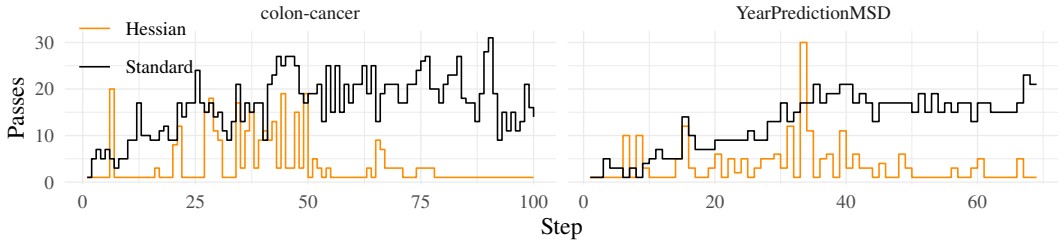

Figure 2: Number of passes of coordinate descent along a full regularization path for the *colon-cancer* ($n = 62$, $p = 2\,000$) and *YearPredictionMSD* ($n = 463\,715$, $p = 90$) data sets, using either Hessian warm starts (7) or standard warm starts (the solution from the previous step).

### 3.3.3  General Loss Functions

We now venture beyond the standard lasso and consider loss functions of the form

$$f(\beta; X) = \sum_{i=1}^{n} f_i(x_i^T \beta) \tag{8}$$

where $f_i$ is convex and twice differentiable. This, for instance, includes logistic, multinomial, and Poisson loss functions. For the strong rule and working set strategy, this extension does not make much of a difference. With the Hessian screening rule, however, the situation is different.

To see this, we start by noting that our method involving the Hessian is really a quadratic Taylor approximation of (1) around a specific point $\beta_0$. For loss functions of the type (8), this approximation is equal to

$$Q(\beta, \beta_0) = f(\beta_0; X) + \sum_{i=1}^{n} \left( x_i^T f_i'(x_i^T \beta_0)(\beta - \beta_0) + \frac{1}{2}(\beta - \beta_0)^T x_i^T f_i''(x_i^T \beta_0) x_i (\beta - \beta_0) \right)$$

$$= \frac{1}{2} \left( \tilde{y}(x_i^T \beta_0) - X\beta \right)^T D\left(w(\beta_0)\right) \left( \tilde{y}(x_i^T \beta_0) - X\beta \right) + C(\beta_0),$$

where $D(w(\beta_0))$ is a diagonal matrix with diagonal entries $w(\beta_0)$ where $w(\beta_0)_i = f''(x_i^T \beta_0)$ and $\tilde{y}(z)_i = f_i'(z)/f_i''(z) - x_i^T \beta_0$, whilst $C(\beta_0)$ is a constant with respect to $\beta$.

Suppose that we are on the lasso path at $\lambda_k$ and want to approximate $c(\lambda_{k+1})$. In this case, we simply replace $f(\beta; X)$ in (1) with $Q(\beta, \hat{\beta}(\lambda_k))$, which leads to the following gradient approximation:

$$c^H(\lambda_{k+1}) = c(\lambda_k) + (\lambda_{k+1} - \lambda_k) X^T D(w) X_{\mathcal{A}_k} (X_{\mathcal{A}_k}^T D(w) X_{\mathcal{A}_k})^{-1} \operatorname{sign}\left( \hat{\beta}(\lambda_k)_{\mathcal{A}_k} \right),$$

where $w = w\left( \hat{\beta}(\lambda_k) \right)$. Unfortunately, we cannot use Algorithm 1 to update $X_{\mathcal{A}_k}^T D(w) X_{\mathcal{A}_k}$. This means that we are forced to either update the Hessian directly at each step, which can be computationally demanding when $|\mathcal{A}_k|$ is large and inefficient when $X$ is very sparse, or to approximate

$D(w)$ with an upper bound. In logistic regression, for instance, we can use ¼ as such a bound, which also means that we once again can use Algorithm 1.

In our experiments, we have employed the following heuristic to decide whether to use an upper bound or compute the full Hessian in these cases: we use full updates at each step if $\text{sparsity}(X)n/\max\{n,p\} < 10^{-3}$ and the upper bound otherwise.

### 3.3.4 Reducing the Impact of KKT Checks

The Hessian Screening Rule is heuristic, which means there may be violations. This necessitates that we verify the KKT conditions after having reached convergence for the screened set of predictors, and add predictors back into the working set for which these checks fail. When the screened set is small relative to $p$, the cost of optimization is often in large part consumed by these checks. Running these checks for the full set of predictors always needs to be done once, but if there are violations during this step, then we need repeat this check, which is best avoided. Here we describe two methods to tackle this issue.

We employ a procedure equivalent to the one used in Tibshirani et al. [10] for the working set strategy: we first check the KKT conditions for the set of predictors singled out by the strong rule and then, if there are no violations in that set, check the full set of predictors for violations. This works well because the strong rule is conservative—violations are rare—which means that we seldom need to run the KKT checks for the entire set more than once.

If we, in spite of the augmentation of the rule, run into violations when checking the full set of predictors, that is, when the strong rule fails to capture the active set, then we can still avoid repeating the full KKT check by relying on Gap Safe screening: after having run the KKT checks and have failed to converge, we screen the set of predictors using the Gap Safe rule. Because this is a safe rule, we can be sure that the predictors we discard will be inactive, which means that we will not need to include them in our upcoming KKT checks. Because Gap Safe screening and the KKT checks rely on exactly the same quantity—the correlation vector–we can do so at marginal extra cost. To see how this works, we now briefly introduce Gap Safe screening. For details, please see Fercoq, Gramfort, and Salmon [6].

For the ordinary lasso ($\ell_1$-regularized least squares), the primal (1) is $P(\beta) = \frac{1}{2}\|y - X\beta\|_2^2 + \lambda\|\beta\|_1$ and the corresponding dual is

$$D(\theta) = \frac{1}{2}\|y\|_2^2 - \frac{\lambda^2}{2}\left\|\theta - \frac{y}{\lambda}\right\|_2^2 \qquad (9)$$

subject to $\|X^T\theta\|_\infty \leq 1$. The duality gap is then $G(\beta, \theta) = P(\beta) - D(\theta)$ and the relation between the primal and dual problems is given by $y = \lambda\hat{\theta} + X\hat{\beta}$, where $\hat{\theta}$ is the maximizer to the dual problem (9). In order to use Gap Safe screening, we need a feasible dual point, which can be obtained via dual point scaling, taking $\theta = (y - X\beta)/\max\left(\lambda, \|X^T(y - X\beta)\|_\infty\right)$. The Gap Safe screening rule then discards the $j$th feature if $|x_j^T\theta| < 1 - \|x_j\|_2\sqrt{2G(\beta,\theta)/\lambda^2}$. Since we have computed $X^T(y - X\beta)$ as part of the KKT checks, we can perform Gap Safe screening at an additional (and marginal) cost amounting to $O(n) + O(p)$.

Since this augmentation benefits the working set strategy too, we adopt it in our implementation of this method as well. To avoid ambiguity, we call this version working+. Note that this makes the working set strategy quite similar to Blitz. In Appendix F.8 we show the benefit of adding this type of screening.

### 3.3.5 Final Algorithm

The Hessian screening method is presented in full in Algorithm 2 (Appendix B).

**Lemma 3.4.** *Let $\beta \in \mathbb{R}^{p\times m}$ be the output of Algorithm 2 for a path of length $m$ and convergence threshold $\varepsilon > 0$. For each step $k$ along the path and corresponding solution $\beta^{(k)} \in \mathbb{R}^p$, there is a dual-feasible point $\theta^{(k)}$ such that $G(\beta^{(k)}, \theta^{(k)}) < \zeta\varepsilon$.*

*Proof.* First note that Gap safe screening [7, Theorem 6] ensures that $\mathcal{G} \supseteq \mathcal{A}_k$. Next, note that the algorithm guarantees that the working set, $\mathcal{W}$, grows with each iteration until $|x_j^T r| < \lambda_k$ for all

$j \in \mathcal{G} \setminus \mathcal{W}$, at which point

$$\max\left(\lambda_k, \|X_{\mathcal{W}}^T(y - X_{\mathcal{W}}\beta_{\mathcal{W}}^{(k)})\|_\infty\right) = \max\left(\lambda_k, \|X_{\mathcal{G}}^T(y - X_{\mathcal{G}}\beta_{\mathcal{G}}^{(k)})\|_\infty\right).$$

At this iteration, convergence at line 2, for the subproblem $(X_{\mathcal{W}}, y)$, guarantees convergence for the full problem, $(X, y)$, since

$$\theta^{(k)} = \frac{y - X_{\mathcal{W}}\beta_{\mathcal{W}}^{(k)}}{\max\left(\lambda_k, \|X_{\mathcal{W}}^T(y - X_{\mathcal{W}}\beta_{\mathcal{W}}^{(k)})\|_\infty\right)}$$

is dual-feasible for the full problem. $\qquad\square$

### 3.3.6 Extensions

**Approximate Homotopy**  In addition to improved screening and warm starts, the Hessian also allows us to construct the regularization path adaptively via approximate homotopy [19]. In brief, the Hessian screening rule allows us to choose the next $\lambda$ along the path adaptively, in effect distributing the grid of $\lambda$s to better approach the exact (homotopy) solution for the lasso, avoiding the otherwise heuristic choice, which can be inappropriate for some data sets.

**Elastic Net**  Our method can be extended to the elastic net [20], which corresponds to adding a quadratic penalty $\phi\|\beta\|_2^2/2$ to (1). The Hessian now takes the form $X_{\mathcal{A}}^T X_{\mathcal{A}} + \phi I$. Loosely speaking, the addition of this term makes the problem "more" quadratic, which in turn improves both the accuracy and stability of the screening and warm starts we use in our method. As far as we know, however, there is unfortunately no way to update the inverse of the Hessian efficiently in the case of the elastic net. More research in this area would be welcome.

## 4 Experiments

Throughout the following experiments, we scale and center predictors with the mean and uncorrected sample standard deviation respectively. For the lasso, we also center the response vector, $y$, with the mean.

To construct the regularization path, we adopt the default settings from `glmnet`: we use a log-spaced path of 100 $\lambda$ values from $\lambda_{\max}$ to $\xi\lambda_{\max}$, where $\xi = 10^{-2}$ if $p > n$ and $10^{-4}$ otherwise. We stop the path whenever the deviance ratio, $1 - \text{dev}/\text{dev}_{\text{null}}$, reaches 0.999 or the fractional decrease in deviance is less than $10^{-5}$. Finally, we also stop the path whenever the number of coefficients ever to be active predictors exceeds $p$.

We compare our method against working+ (the modified version of the working set strategy from Tibshirani et al. [10]), Celer [15], and Blitz [14]. We initially also ran our comparisons against EDPP [9], the Gap Safe rule [6], and Dynamic Sasvi [8] too, yet these methods performed so poorly that we omit the results in the main part of this work. The interested reader may nevertheless consult Appendix F.6 where results from simulated data has been included for these methods too.

We use cyclical coordinate descent with shuffling and consider the model to converge when the duality gap $G(\beta, \theta) \le \varepsilon\zeta$, where we take $\zeta$ to be $\|y\|_2^2$ when fitting the ordinary lasso, and $n \log 2$ when fitting $\ell_1$-regularized logistic regression. Unless specified, we let $\varepsilon = 10^{-4}$. These settings are standard settings and, for instance, resemble the defaults used in Celer. For all of the experiments, we employ the line search algorithm used in Blitz[4].

The code used in these experiments was, for every method, programmed in C++ using the Armadillo library [21, 22] and organized as an R package via Rcpp [23]. We used the renv package [24] to maintain dependencies. The source code, including a Singularity [25] container and its recipe for reproducing the results, are available at `https://github.com/jolars/HessianScreening`. Additional details of the computational setup are provided in Appendix D.

---

[4]Without the line search, all of the tested methods ran into convergence issues, particularly for the high-correlation setting and logistic regression.

## 4.1 Simulated Data

Let $X \in \mathbb{R}^{n \times p}$, $\beta \in \mathbb{R}^p$, and $y \in \mathbb{R}^n$ be the predictor matrix, coefficient vector, and response vector respectively. We draw the rows of the predictor matrix independently and identically distributed from $\mathcal{N}(0, \Sigma)$ and generate the response from $\mathcal{N}(X\beta, \sigma^2 I)$ with $\sigma^2 = \beta^T \Sigma \beta / \text{SNR}$, where SNR is the signal-to-noise ratio. We set $s$ coefficients, equally spaced throughout the coefficient vector, to 1 and the rest to zero.

In our simulations, we consider two scenarios: a low-dimensional scenario and a high-dimensional scenario. In the former, we set $n = 10\,000$, $p = 100$, $s = 5$, and the SNR to 1. In the high-dimensional scenario, we take $n = 400$, $p = 40\,000$, $s = 20$, and set the SNR to 2. These SNR values are inspired by the discussion in Hastie, Tibshirani, and Tibshirani [26] and intend to cover the middle-ground in terms of signal strength. We run our simulations for 20 iterations.

From Figure 3, it is clear that the Hessian screening rule performs best, taking the least time in every setting examined. The difference is largest for the high-correlation context in the low-dimensional setting and otherwise roughly the same across levels of correlation.

The differences between the other methods are on average small, with the working+ strategy performing slightly better in the $p > n$ scenario. Celer and Blitz perform largely on par with one another, although Celer sees an improvement in a few of the experiments, for instance in logistic regression when $p > n$.

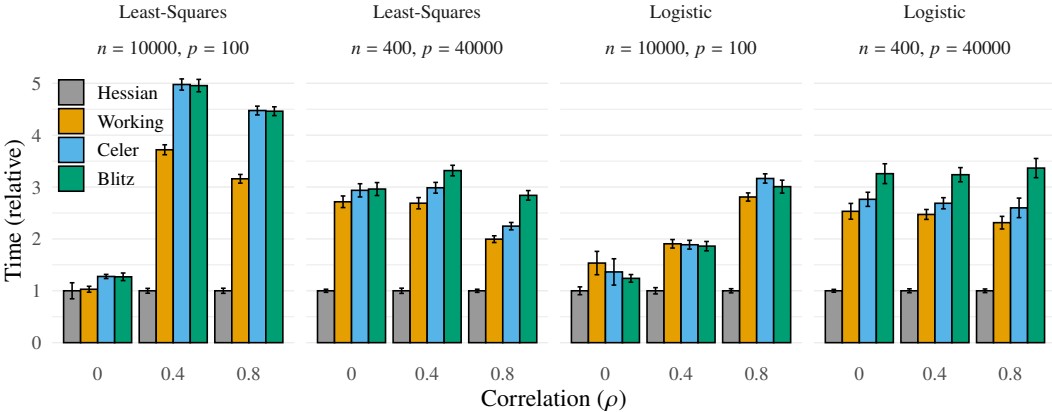

Figure 3: Time to fit a full regularization path for $\ell_1$-regularized least-squares and logistic regression to a design with $n$ observations, $p$ predictors, and pairwise correlation between predictors of $\rho$. Time is relative to the minimal mean time in each group. The error bars represent ordinary 95% confidence intervals around the mean.

## 4.2 Real Data

In this section, we conduct experiments on real data sets. We run 20 iterations for the smaller data sets studied and three for the larger ones. For information on the sources of these data sets, please see Appendix E. For more detailed results of these experiments, please see Appendix F.5.

Starting with the case of $\ell_1$-regularized least-squares regression, we observe that the Hessian screening rule performs best for all five data sets tested here (Table 1), in all but one instance taking less than half the time compared to the runner-up, which in each case is the working+ strategy. The difference is particularly large for the YearPredictionMSD and e2006-tfidf data sets.

In the case of $\ell_1$-regularized logistic regression, the Hessian method again performs best for most of the examined data sets, for instance completing the regularization path for the madelon data set around five times faster than the working+ strategy. The exception is the arcene data set, for which the working+ strategy performs best out of the four methods.

We have provided additional results related to the effectiveness of our method in Appendix F.

Table 1: Average time to fit a full regularization path of $\ell_1$-regularized least-squares and logistic regression to real data sets. Density represents the fraction of non-zero entries in $X$. Density and time values are rounded to two and three significant figures respectively.

| | | | | | Time (s) | | | |
|---|---|---|---|---|---|---|---|---|
| Data Set | $n$ | $p$ | Density | Loss | Hessian | Working | Blitz | Celer |
| bcTCGA | 536 | 17 322 | 1 | Least-Squares | 3.00 | 7.67 | 11.7 | 10.6 |
| e2006-log1p | 16 087 | 4 272 227 | $1.4 \times 10^{-3}$ | Least-Squares | 205 | 438 | 756 | 835 |
| e2006-tfidf | 16 087 | 150 360 | $8.3 \times 10^{-3}$ | Least-Squares | 14.3 | 143 | 277 | 335 |
| scheetz | 120 | 18 975 | 1 | Least-Squares | 0.369 | 0.643 | 0.706 | 0.801 |
| YearPredictionMSD | 463 715 | 90 | 1 | Least-Squares | 78.8 | 541 | 706 | 712 |
| arcene | 100 | 10 000 | $5.4 \times 10^{-1}$ | Logistic | 4.35 | 3.27 | 4.42 | 3.99 |
| colon-cancer | 62 | 2000 | 1 | Logistic | 0.0542 | 0.134 | 0.177 | 0.169 |
| duke-breast-cancer | 44 | 7129 | 1 | Logistic | 0.111 | 0.210 | 0.251 | 0.262 |
| ijcnn1 | 35 000 | 22 | 1 | Logistic | 0.939 | 5.53 | 4.68 | 3.50 |
| madelon | 2000 | 500 | 1 | Logistic | 48.2 | 232 | 240 | 247 |
| news20 | 19 996 | 1 355 191 | $3.4 \times 10^{-4}$ | Logistic | 1290 | 1620 | 2230 | 2170 |
| rcv1 | 20 242 | 47 236 | $1.6 \times 10^{-3}$ | Logistic | 132 | 266 | 384 | 378 |

## 5 Discussion

We have presented the Hessian Screening Rule: a new heuristic predictor screening rule for $\ell_1$-regularized generalized linear models. We have shown that our screening rule offers large performance improvements over competing methods, both in simulated experiments but also in the majority of the real data sets that we study here. The improved performance of the rule appears to come not only from improved effectiveness in screening, particularly in the high-correlation setting, but also from the much-improved warm starts, which enables our method to dominate in the $n \gg p$ setting. Note that although we have focused on $\ell_1$-regularized least-squares and logistic regression here, our rule is applicable to any composite objective for which the differentiable part is twice-differentiable.

One limitation of our method is that it consumes more memory than its competitors owing to the storage of the Hessian and its inverse. This cost may become prohibitive for cases when $\min\{n, p\}$ is large. In these situations the next-best choice may instead be the working set strategy. Note also that we, in this paper, focus entirely on the lasso *path*. The Hessian Screening Rule is a sequential rule and may therefore not prove optimal when solving for a single $\lambda$, in which case a dynamic strategy such as Celer and Blitz likely performs better.

With respect to the relative performance of the working set strategy, Celer, and Blitz, we note that our results deviate somewhat from previous comparisons [15, 14]. We speculate that these differences might arise from the fact that we have used equivalent implementations for all of the methods and from the modification that we have used for the working set strategy.

Many avenues remain to be explored in the context of Hessian-based screening rules and algorithms, such as developing more efficient methods for updating of the Hessian matrix for non-least-squares objectives, such as logistic regression and using second-order information to further improve the optimization method used. Other interesting directions also include adapting the rules to more complicated regularization problems, such as the fused lasso [27], SLOPE [28], SCAD [29], and MCP [30]. Although the latter two of these are non-convex problems, they are locally convex for intervals of the regularization path [31], which enables the use of our method. Adapting the method for use in batch stochastic gradient descent would also be an interesting topic for further study, for instance by using methods such as the ones outlined in Asar et al. [32] to ensure that the Hessian remains positive definite.

Finally, we do not expect there to be any negative societal consequences of our work given that it is aimed solely at improving the performance of an optimization method.

## Acknowledgments and Disclosure of Funding

We would like to thank Małgorzata Bogdan for valuable comments. This work was funded by the Swedish Research Council through grant agreement no. 2020-05081 and no. 2018-01726. The computations were enabled by resources provided by LUNARC. The results shown here are in part based upon data generated by the TCGA Research Network: https://www.cancer.gov/tcga.

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
