# Supplement to *The Hessian Screening Rule*

**Johan Larsson**
Department of Statistics
Lund University
johan.larsson@stat.lu.se

**Jonas Wallin**
Department of Statistics
Lund University
jonas.wallin@stat.lu.se

## A   Proofs

### A.1   Proof of Theorem 1

It suffices to verify that the KKT conditions hold for $\hat{\beta}^{\lambda^*}(\lambda)$, i.e. that $\mathbf{0}$ is in the subdifferential. By (ii) it follows that the indices $\mathcal{A}_{\lambda^*}^c$ in the subdifferential contain zero. That leaves us only to show that $\nabla f\big(\hat{\beta}^{\lambda^*}(\lambda); X\big)_{\mathcal{A}_{\lambda^*}} = \lambda \operatorname{sign}\big(\hat{\beta}^{\lambda^*}(\lambda)\big)_{\mathcal{A}_{\lambda^*}}$.

$$
\begin{aligned}
&\nabla f\big(\hat{\beta}^{\lambda^*}(\lambda); X\big)_{\mathcal{A}_{\lambda^*}} \\
&= X_{\mathcal{A}_{\lambda^*}}^T \big( y - X_{\mathcal{A}_{\lambda^*}} \hat{\beta}^{\lambda^*}(\lambda)_{\mathcal{A}_{\lambda^*}} \big) \\
&= X_{\mathcal{A}_{\lambda^*}}^T \left( y - X_{\mathcal{A}_{\lambda^*}} \beta(\lambda^*)_{\mathcal{A}_{\lambda^*}} - (\lambda^* - \lambda) X_{\mathcal{A}_{\lambda^*}} \big( X_{\mathcal{A}_{\lambda^*}}^T X_{\mathcal{A}_{\lambda^*}} \big)^{-1} \operatorname{sign} \beta(\lambda^*)_{\mathcal{A}_{\lambda^*}} \right) \\
&= \nabla f\big(\hat{\beta}^{\lambda^*}(\lambda^*)\big)_{\mathcal{A}_{\lambda^*}} - (\lambda^* - \lambda) \operatorname{sign} \hat{\beta}(\lambda^*)_{\mathcal{A}_{\lambda^*}} \\
&= \lambda \operatorname{sign} \hat{\beta}(\lambda^*)_{\mathcal{A}_{\lambda^*}},
\end{aligned}
$$

which by (i) equals $\lambda \operatorname{sign}(\hat{\beta}^{\lambda^*}(\lambda))_{\mathcal{A}_{\lambda^*}}$.

## B   Algorithms

In this section we present the algorithms for efficiently updating the Hessian and its inverse (Algorithm 1) and the full algorithm for the Hessian screening method (Algorithm 2).

## C   Singular or Ill-Conditioned Hessians

In this section, we discuss situations in which the Hessian is singular or ill-conditioned and propose remedies for these situations.

Inversion of the Hessian demands that the null space corresponding to the active predictors $\mathcal{A}_\lambda$ contains only the zero vector, which typically holds when the columns of $X$ are in general position, such as in the case of data simulated from continuous distributions. It is not, however, generally the case with discrete-valued data, particularly not in when $p \gg n$. In Lemma C.1, we formalize this point.

**Lemma C.1.** *Suppose that we have $e \in \mathbb{R}^p$ such that $Xe = 0$. Let $\hat{\beta}(\lambda)$ be the solution to the primal problem (1) and $\mathcal{E} = \{i : e_i \neq 0\}$; then $|\hat{\beta}(\lambda)_{\mathcal{E}}| > 0$ only if there exists a $z \in \mathbb{R}^p$ where $z_{\mathcal{E}} \in \{-1, 1\}^{|\mathcal{E}|}$ such that $z^T e = 0$.*

36th Conference on Neural Information Processing Systems (NeurIPS 2022).

**Algorithm 1** This algorithm provides computationally efficient updates for the inverse of the Hessian. Note the slight abuse of notation here in that $\mathcal{E}$ is used both for $X$ and $Q$. It is implicitly understood that $Q_{\mathcal{E}\mathcal{E}}$ is the sub-matrix of $Q$ that corresponds to the columns $\mathcal{E}$ of $X$.

---

**Input:** $X, H = X_{\mathcal{A}}^T X_{\mathcal{A}}, Q := H^{-1}, \mathcal{A}, \mathcal{B}$
  $\mathcal{C} := \mathcal{A} \setminus \mathcal{B}$
  $\mathcal{D} := \mathcal{B} \setminus \mathcal{A}$
  **if** $\mathcal{C} \neq \varnothing$ **then**
    $\mathcal{E} := \mathcal{A} \cap \mathcal{B}$
    $Q := Q_{\mathcal{E}\mathcal{E}} - Q_{\mathcal{E}\mathcal{E}^c} Q_{\mathcal{E}^c\mathcal{E}^c}^{-1} Q_{\mathcal{E}\mathcal{E}^c}^T$
    $\mathcal{A} := \mathcal{E}$
  **end if**
  **if** $\mathcal{D} \neq \varnothing$ **then**
    $S := X_{\mathcal{D}}^T X_{\mathcal{D}} - X_{\mathcal{D}}^T X_{\mathcal{A}} Q X_{\mathcal{A}}^T X_{\mathcal{D}}$
    $Q := \begin{bmatrix} Q + Q X_{\mathcal{A}}^T X_{\mathcal{D}} S^{-1} X_{\mathcal{D}}^T X_{\mathcal{A}} Q & -Q X_{\mathcal{A}}^T X_{\mathcal{D}} S^{-1} \\ -S^{-1} X_{\mathcal{D}}^T X_{\mathcal{A}} Q & S^{-1} \end{bmatrix}$
  **end if**
  Return $H^*$

---

**Algorithm 2** The Hessian screening method for the ordinary least-squares lasso

---

**Input:** $X \in \mathbb{R}^{n \times p}, y \in \mathbb{R}^n, \lambda \in \{\mathbb{R}_+^m : \lambda_1 = \lambda_{\max}, \lambda_1 > \lambda_2 > \cdots > \lambda_m\}, \varepsilon > 0$
**Initalize:** $k \leftarrow 1, \beta^{(0)} \leftarrow 0, \zeta \leftarrow \|y\|_2^2, \mathcal{W} \leftarrow \varnothing, \mathcal{A} \leftarrow \varnothing, \mathcal{S} \leftarrow \varnothing, \mathcal{G} \leftarrow \{1, 2, \ldots, p\}$

1:  **while** $k \leq m$ **do**
2:    $\beta_{\mathcal{W}}^{(k)} \leftarrow \{\beta \in \mathbb{R}^{|\mathcal{W}|} : G(\beta, (y - X_{\mathcal{W}}\beta)/\max(\lambda_k, \|X_{\mathcal{W}}^T(y - X_{\mathcal{W}}\beta)\|_\infty)) < \zeta\varepsilon\}$
3:    $\beta_{\mathcal{W}^c}^{(k)} \leftarrow 0$
4:    $\mathcal{A} \leftarrow \{j : \beta_j \neq 0\}$
5:    $r \leftarrow y - X_{\mathcal{W}}\beta_{\mathcal{W}}^{(k)}$
6:    $\mathcal{V} \leftarrow \{j \in \mathcal{S} \setminus \mathcal{W} : |x_j^T r| \geq \lambda_k\}$          ▷ *Check for violations in Strong set*
7:    **if** $\mathcal{V} = \varnothing$ **then**
8:      $\theta \leftarrow r/\max\left(\lambda_k, \|X_{\mathcal{G}}^T r\|_\infty\right)$         ▷ *Compute dual-feasible point*
9:      **if** $G(\beta^{(k)}, \theta) < \varepsilon\zeta$ **then**
10:        Update $H$ and $H^{-1}$ via Algorithm 1
11:        $\mathcal{W} \leftarrow \{j : |\tilde{c}^H(\lambda_{k+1})| < \lambda_{k+1}\} \cup \mathcal{A}$        ▷ *Hessian rule screening*
12:        $\mathcal{S} \leftarrow \{j : |\tilde{c}^S(\lambda_{k+1})| < \lambda_{k+1}\}$           ▷ *Strong rule screening*
13:        Initialize $\beta_{\mathcal{A}}^{(k+1)}$ using (7)            ▷ *Hessian warm start*
14:        $\mathcal{G} \leftarrow \{1, 2, \ldots, p\}$                 ▷ *Reset Gap-Safe set*
15:        $k \leftarrow k + 1$                    ▷ *Move to next step on path*
16:      **else**
17:        $\mathcal{G} \leftarrow \left\{j \in \mathcal{G} : |x_j^T \theta| \geq 1 - \|x_j\|_2 \sqrt{2G(\beta^{(k)}, \theta)/\lambda_k^2}\right\}$   ▷ *Gap-Safe screening*
18:        $\mathcal{V} \leftarrow \{j \in \mathcal{G} \setminus (\mathcal{S} \cup \mathcal{W}) : |x_j^T r| \geq \lambda_k\}$   ▷ *Check for violations in Gap-Safe set*
19:        $\mathcal{W} \leftarrow \mathcal{W} \cap \mathcal{G}$
20:        $\mathcal{S} \leftarrow \mathcal{S} \cap \mathcal{G}$
21:      **end if**
22:    **end if**
23:    $\mathcal{W} \leftarrow \mathcal{W} \cup \mathcal{V}$          ▷ *Augment working set with violating predictors*
24: **end while**
25: **return** $\beta$

*Proof.* $\sum_{j \in \mathcal{E}} x_j e_j = 0$ by assumption. Then, since $\hat{\beta}(\lambda)$ is the solution to the primal problem, it follows that $x_j^T \nabla f(X\beta) = \text{sign}(\beta_j)\lambda$ for all $j \in \mathcal{E}$. Hence

$$\sum_{j \in \mathcal{E}} x_j^T \nabla f(X\beta) e_j = \sum_{j \in \mathcal{E}} \text{sign}(\beta_j)\lambda e_j = \lambda \sum_{j \in \mathcal{E}} \text{sign}(\beta_j) e_j = 0$$

and $z_{\mathcal{E}^C} = 0$, $z_{\mathcal{E}} = \text{sign}(\beta_{\mathcal{E}})$. □

In our opinion, the most salient feature of this result is that if all predictors in $\mathcal{E}$ except $i$ are known to be active, then predictor $i$ is active iff $e_i = \sum_{j \in \mathcal{E} \setminus i} \pm e_j$. If the columns of $X$ are independent and normally distributed, this cannot occur and hence one will never see a null space in $X_{\mathcal{A}}$. Yet if $X_{ij} \in \{0, 1\}$, one should expect the null space to be non-empty frequently. A simple instance of this occurs when the columns of $X$ are duplicates, in which case $|e| = 2$.

Duplicated predictors are fortunately easy to handle since they enter the model simultaneously. And we have, in our program, implemented measures that deal efficiently with this issue by dropping them from the solution after fitting and adjust $\hat{\beta}$ accordingly.

Dealing with the presence of rank-deficiencies due to the existence of linear combinations among the predictors is more challenging. In the work for this paper, we developed a strategy to deal with this issue directly by identifying such linear combinations through spectral decompositions. During our experiments, however, we discovered that this method often runs into numerical issues that require other modifications that invalidate its potential. We have therefore opted for a different strategy.

To deal with singularities and ill-conditioned Hessian matrices, we instead use preconditioning. At step $k$, we form the spectral decomposition

$$H_{\mathcal{A}_k} = Q\Lambda Q^T.$$

Then, if $\min_i \big(\text{diag}(\Lambda)\big) < \alpha$, we add a factor $\alpha$ to the diagonal of $H_{\mathcal{A}_k}$. Then we substitute

$$\hat{H}_{\mathcal{A}_k}^{-1} = Q^T (I\alpha + \Lambda)^{-1} Q$$

for the true Hessian inverse. An analogous approach is taken when updating the Hessian incrementally as in Algorithm 1. In our experiments, we have set $\alpha := n10^{-4}$.

## D   Computational Setup Details

The computer used to run the experiments had the following specifications:

**CPU**  Intel i7-10510U @ 1.80Ghz (4 cores)

**Memory**  64 GB (3.2 GB/core)

**OS**  Fedora 36

**Compiler**  GNU GCC compiler v9.3.0, C++17

**BLAS/LAPACK**  OpenBLAS v0.3.8

**R version**  4.1.3

## E   Real Data Sets

All of the data sets except *arcene*, *scheetz*, and *bc_tcga* werew retrieved from https://www.csie.ntu.edu.tw/~cjlin/libsvmtools/datasets/ [1, 2]. arcene was retrieved from https://archive.ics.uci.edu/ml/datasets/Arcene [3, 4] and scheetz and bc_tcga from https://myweb.uiowa.edu/pbreheny [5]. Their original sources have been listed in Table 1. In each case where it is available we use the training partition of the data set and otherwise the full data set.

## F   Additional Results

In this section, we present additional results related to the performance of the Hessian Screening Rule.

Table 1: Source for the real data sets used in our experiments.

| Dataset | Sources |
| --- | --- |
| arcene | Guyon et al. [3] and Dua and Graff [4] |
| bcTCGA | National Cancer Institute [6] |
| colon-cancer | Alon et al. [7] |
| duke-breast-cancer | West et al. [8] |
| e2006-log1p | Kogan et al. [9] |
| e2006-tfidf | Kogan et al. [9] |
| ijcnn1 | Prokhorov [10] |
| madelon | Guyon et al. [3] |
| news20 | Keerthi and DeCoste [11] |
| rcv1 | Lewis et al. [12] |
| scheetz | Scheetz et al. [13] |
| YearPredictionMSD | Bertin-Mahieux et al. [14] and Dua and Graff [4] |

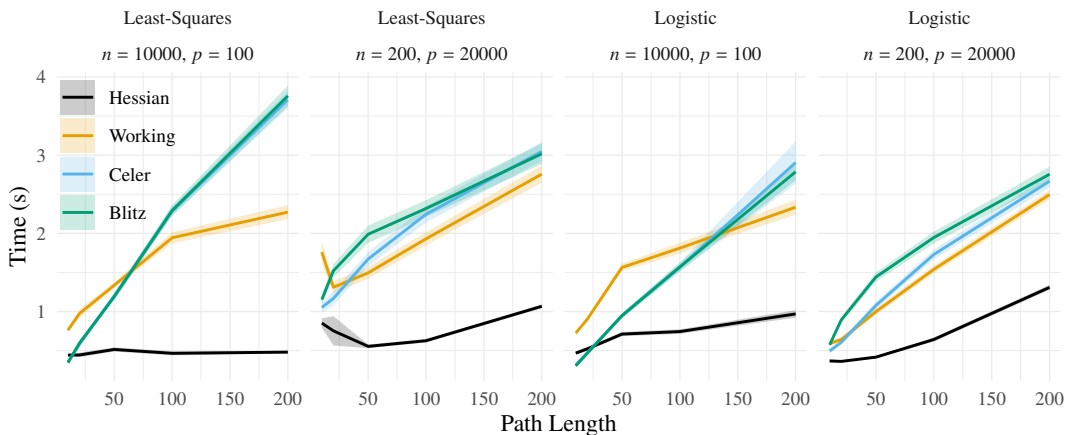

Figure 1: The time in seconds required to fit a full regularization path with length given on the x axis.

## F.1 Path Length

Using the same setup as in Section 4 but with $n = 200$, $p = 20\,000$ for the high-dimensional setting, we again benchmark the time required to fit a full regularization path using the different methods studied in this paper. The results (Figure 1) show that the Hessian Screening Method out-performs the studied alternatives except for the low-dimensional situation and a path length of 10 $\lambda$s. The results demonstrate that our method pays a much smaller price for increased path resolution compared to the other methods but that the increased marginal costs of updating the Hessian may make the method less appealing in this case.

## F.2 Convergence Tolerance

To better understand if and how the stopping threshold used in the solver affects the performance of the various methods we test, we conduct simulations where we vary the tolerance, keeping the remaining parameters constant. We use the same situation as in the high-dimensional scenario (see Section 4) but use $n = 200$, $p = 20\,000$. We run the experiment for tolerances $10^{-3}, 10^{-4}, 10^{-5}$, and $10^{-6}$. The results (Figure 2) indicate that the choice of stopping threshold has some importance for convergence time but that the gap between our method and the alternatives tested never disappears.

## F.3 The Benefit of Augmenting Heuristic Methods with Gap Safe Screening

To study the effectiveness of augmenting the Hessian Screening and working methods with a gap-safe check, we conduct experiments using the high-dimensional setup in Section 4 but with $n = 200$ and

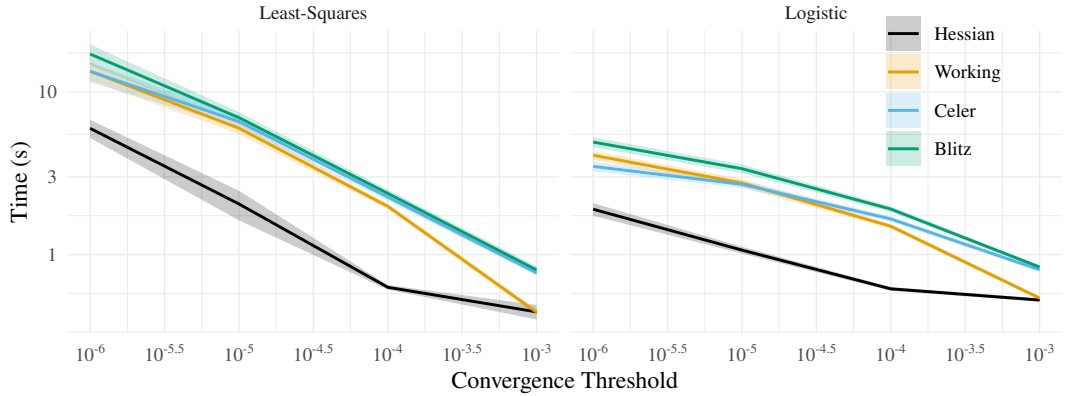

Figure 2: Time required to fit a full regularization path for the high-dimensional scenario setup in Section 4 for both $\ell_1$-regularized least-squares and logistic regression, with $n = 200$ and $p = 20\,000$. Both the x and y axis are on a $\log_{10}$ scale.

$p = 20\,000$, either enabling this augmentation or disabling it. We also vary the level of correlation, $\rho$. Each combination is benchmarked across 20 iterations.

The results indicate that the addition of gap safe screening makes a definite, albeit modest, contribution to the performance of the methods, particularly in the case of the working strategy, which is to be expected given that the working strategy typically runs more KKT checks that the Hessian method does since it causes many more violations.

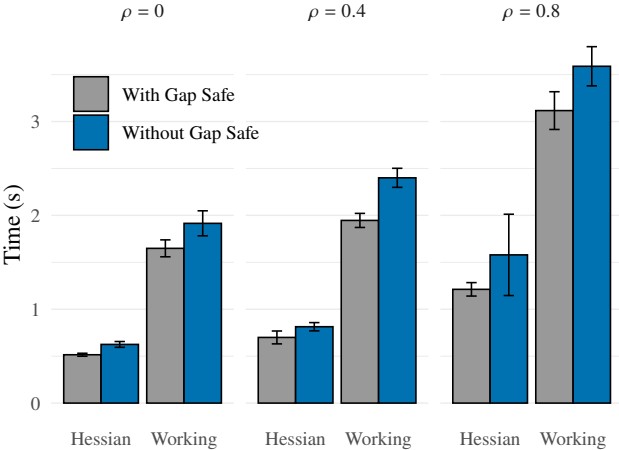

Figure 3: Average time in seconds required to fit a full regularization path for the high-dimensional scenario setup in Section 4 for $\ell_1$-regularized least-squares regression, with $n = 200$ and $p = 20\,000$, using the Hessian and working set methods with or without the addition of Gap Safe screening. The bars represent ordinary 95% confidence intervals.

## F.4 Effectiveness and Violations

To study the effectiveness of the screening rule, we conduct as experiment using the setup in Section 4 but with $n = 200$ and $p = 20\,000$. We run 20 iterations and average the number of screened predictors as well as violations across the entire path.

Looking at the effectiveness of the screening rules, we see that the Hessian screening rule performs as desired for both $\ell_1$-regularized least-squares and logistic regression (Figure 4), leading to a screened set that lies very close to the true size. In particular, the rule works much better than all alternatives in the case of high correlation,

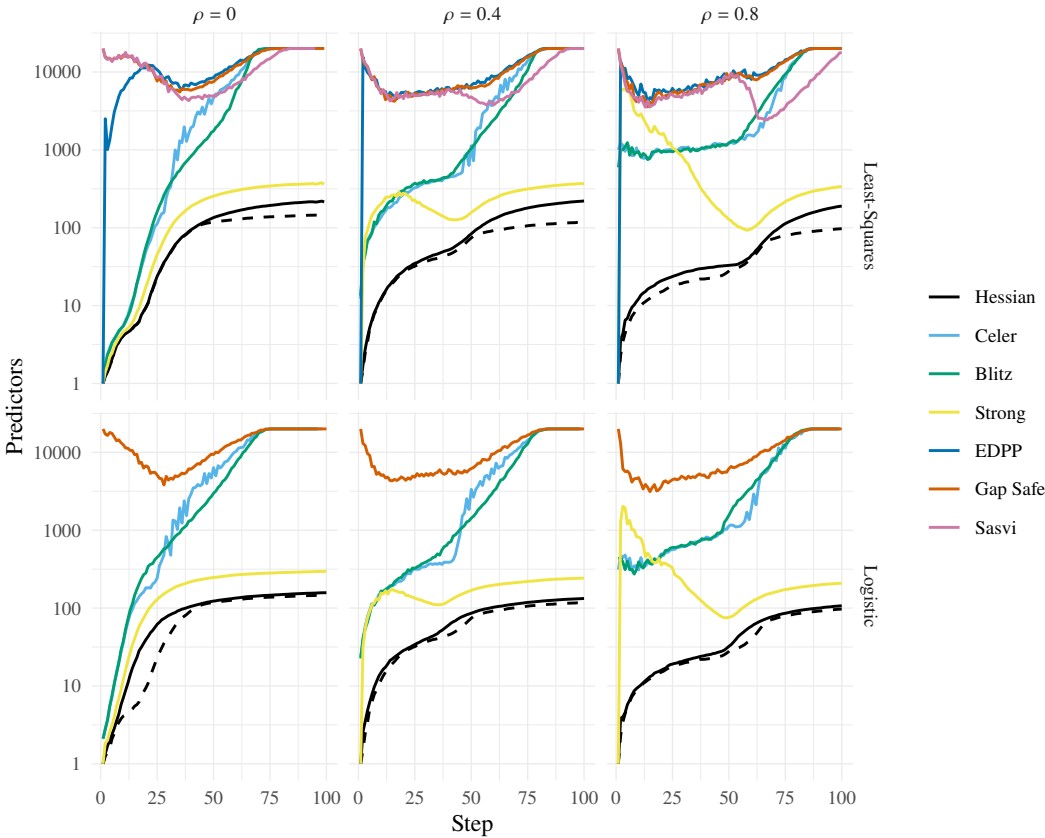

Figure 4: The number of predictors screened (included) for each given screening rule, as well as the minimum number of active predictors at each step as a dashed line. The values are averaged over 20 repetitions in each condition. Note that the y-axis is on a $\log_{10}$ scale.

In Table 2, we show the average numbers of screened (included) predictors and violations for the heuristic screening rules across the path. We note, first, that EDPP never lead to any violations and that the Strong rule only did so once throughout the experiments. The Hessian rule, on the other hand, leads to more violations, particularly when there is high correlation. On the other hand, the Hessian screening rule successfully discards many more predictors than the other two rules do. And because the Hessian method always checks for violations in the strong rule set first, which is demonstrably conservative, these violations are of little importance in practice.

## F.5 Detailed Results on Real Data

In Table 3 we show Table 1 with additional detail, including confidence intervals and higher figure resolutions. Please see Section 4 for commentary on these results, where they have been covered in full.

## F.6 Additional Results on Simulated Data

In Figure 5, we show results for the ordinary least-squares lasso for the Sasvi, Gap Safe, and EDPP methods, which were not included in the main paper.

## F.7 Gamma

In this section we present the results of experiments targeting $\gamma$, the parameter for the Hessian rule that controls how much of the unit bound (used in the Strong Rule) that is included in the correlation vector estimate from the Hessian rule.

Table 2: Numbers of screened predictors and violations averaged over the entire path and 20 iterations for simulated data with $n = 20\,000$ $p = 200$ and correlation level equal to $\rho$.

| Model | $\rho$ | Method | Screened | Violations |
|---|---|---|---|---|
| Least-Squares | 0 | Hessian | 112 | 0.081 |
| Least-Squares | 0 | Strong | 203 | 0 |
| Least-Squares | 0 | EDPP | 11 928 | 0 |
| Least-Squares | 0.4 | Hessian | 103 | 0.099 |
| Least-Squares | 0.4 | Strong | 238 | 0 |
| Least-Squares | 0.4 | EDPP | 10 561 | 0 |
| Least-Squares | 0.8 | Hessian | 66 | 0.37 |
| Least-Squares | 0.8 | Strong | 897 | 0.0010 |
| Least-Squares | 0.8 | EDPP | 10 652 | 0 |
| Logistic | 0 | Hessian | 102 | 0.020 |
| Logistic | 0 | Strong | 201 | 0 |
| Logistic | 0.4 | Hessian | 77 | 0.033 |
| Logistic | 0.4 | Strong | 173 | 0 |
| Logistic | 0.8 | Hessian | 49 | 0.051 |
| Logistic | 0.8 | Strong | 297 | 0 |

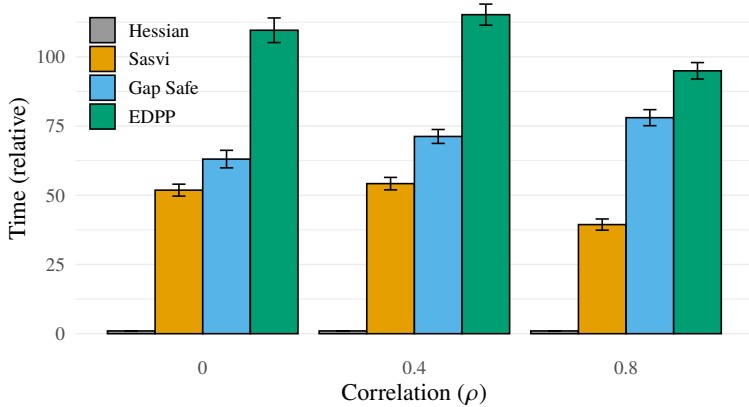

Figure 5: Additional results on simulated data for methods not included in the main article. The results correspond to the ordinary (least-squares) lasso with $n = 400$, $p = 40\,000$ and varying levels of pairwise correlation between predictors, $\rho$.

We run 50 iterations of the high-dimensional setup from Section 4 and measure the number of predictors screened (included) by the Hessian screening rule, the number of violations, and the time taken to fit the full path. We vary $\gamma$ from 0.001 to 0.3.

The results are presented in Figure 6. From the figure it is clear that the number of violations in fact has a slightly negative impact on the speed at which the path is fit. We also see that the number of violations is small considering the dimension of the data set ($p = 40\,000$) and approach zero at $\gamma$ values around 0.1 for the lowest level of correlation, but have yet to reach exactly zero at 0.3 for the highest level of correlation. The size of the screened set increase only marginally as $\gamma$ increases fro 0.001 to 0.01, but eventually increase rapidly at $\gamma$ approaches 0.3. Note, however, that the screened set is still very small relative to the full set of predictors.

### F.8 Ablation Analysis

In this section we report an experiment wherein we study the effects of the various features of the Hessian screening method by incrementally adding them and timing the result.

We add features incrementally in the following order, such that each step includes all of the previous features.

1. Hessian screening

Table 3: Time to fit a full regularization path of $\ell_1$-regularized least-squares and logistic regression to real data sets. Density and time values are rounded to two and four significant figures respectively. The estimates are based on 20 repetitions for arcene, colon-cancer, duke-breast-cancer, and ijcnn1 and three repetitions otherwise. Standard 95% confidence levels are included.

| Dataset | $n$ | $p$ | Density | Loss | Method | Time (s) | 95% CI Lower | 95% CI Upper |
|---|---|---|---|---|---|---|---|---|
| arcene | 100 | 10 000 | $5.4 \times 10^{-1}$ | Logistic | Blitz | 4.42 | 4.39 | 4.45 |
| arcene | 100 | 10 000 | $5.4 \times 10^{-1}$ | Logistic | Celer | 3.99 | 3.98 | 3.99 |
| arcene | 100 | 10 000 | $5.4 \times 10^{-1}$ | Logistic | Hessian | 4.35 | 4.32 | 4.38 |
| arcene | 100 | 10 000 | $5.4 \times 10^{-1}$ | Logistic | Working | 3.27 | 3.25 | 3.28 |
| bcTCGA | 536 | 17 322 | 1 | Least-Squares | Blitz | 11.7 | 11.5 | 11.8 |
| bcTCGA | 536 | 17 322 | 1 | Least-Squares | Celer | 10.6 | 10.5 | 10.7 |
| bcTCGA | 536 | 17 322 | 1 | Least-Squares | Hessian | 3.00 | 2.85 | 3.14 |
| bcTCGA | 536 | 17 322 | 1 | Least-Squares | Working | 7.67 | 7.57 | 7.77 |
| colon-cancer | 62 | 2000 | 1 | Logistic | Blitz | 0.177 | 0.176 | 0.178 |
| colon-cancer | 62 | 2000 | 1 | Logistic | Celer | 0.169 | 0.168 | 0.170 |
| colon-cancer | 62 | 2000 | 1 | Logistic | Hessian | 0.0542 | 0.0534 | 0.0550 |
| colon-cancer | 62 | 2000 | 1 | Logistic | Working | 0.134 | 0.132 | 0.136 |
| duke-breast-cancer | 44 | 7129 | 1 | Logistic | Blitz | 0.251 | 0.248 | 0.253 |
| duke-breast-cancer | 44 | 7129 | 1 | Logistic | Celer | 0.262 | 0.260 | 0.264 |
| duke-breast-cancer | 44 | 7129 | 1 | Logistic | Hessian | 0.111 | 0.110 | 0.112 |
| duke-breast-cancer | 44 | 7129 | 1 | Logistic | Working | 0.210 | 0.209 | 0.212 |
| e2006-log1p | 16 087 | 4 272 227 | $1.4 \times 10^{-3}$ | Least-Squares | Blitz | 756 | 749 | 764 |
| e2006-log1p | 16 087 | 4 272 227 | $1.4 \times 10^{-3}$ | Least-Squares | Celer | 835 | 831 | 839 |
| e2006-log1p | 16 087 | 4 272 227 | $1.4 \times 10^{-3}$ | Least-Squares | Hessian | 205 | 203 | 207 |
| e2006-log1p | 16 087 | 4 272 227 | $1.4 \times 10^{-3}$ | Least-Squares | Working | 438 | 434 | 441 |
| e2006-tfidf | 16 087 | 150 360 | $8.3 \times 10^{-3}$ | Least-Squares | Blitz | 277 | 275 | 280 |
| e2006-tfidf | 16 087 | 150 360 | $8.3 \times 10^{-3}$ | Least-Squares | Celer | 335 | 334 | 337 |
| e2006-tfidf | 16 087 | 150 360 | $8.3 \times 10^{-3}$ | Least-Squares | Hessian | 14.3 | 14.3 | 14.4 |
| e2006-tfidf | 16 087 | 150 360 | $8.3 \times 10^{-3}$ | Least-Squares | Working | 143 | 139 | 146 |
| ijcnn1 | 35 000 | 22 | 1 | Logistic | Blitz | 4.68 | 3.82 | 5.53 |
| ijcnn1 | 35 000 | 22 | 1 | Logistic | Celer | 3.50 | 3.42 | 3.58 |
| ijcnn1 | 35 000 | 22 | 1 | Logistic | Hessian | 0.939 | 0.869 | 1.01 |
| ijcnn1 | 35 000 | 22 | 1 | Logistic | Working | 5.53 | 4.57 | 6.48 |
| madelon | 2000 | 500 | 1 | Logistic | Blitz | 240 | 223 | 258 |
| madelon | 2000 | 500 | 1 | Logistic | Celer | 247 | 243 | 251 |
| madelon | 2000 | 500 | 1 | Logistic | Hessian | 48.2 | 43.2 | 53.1 |
| madelon | 2000 | 500 | 1 | Logistic | Working | 232 | 227 | 238 |
| news20 | 19 996 | 1 355 191 | $3.4 \times 10^{-4}$ | Logistic | Blitz | 2230 | 2230 | 2240 |
| news20 | 19 996 | 1 355 191 | $3.4 \times 10^{-4}$ | Logistic | Celer | 2170 | 2160 | 2180 |
| news20 | 19 996 | 1 355 191 | $3.4 \times 10^{-4}$ | Logistic | Hessian | 1290 | 1290 | 1290 |
| news20 | 19 996 | 1 355 191 | $3.4 \times 10^{-4}$ | Logistic | Working | 1620 | 1610 | 1630 |
| rcv1 | 20 242 | 47 236 | $1.6 \times 10^{-3}$ | Logistic | Blitz | 384 | 380 | 387 |
| rcv1 | 20 242 | 47 236 | $1.6 \times 10^{-3}$ | Logistic | Celer | 378 | 373 | 384 |
| rcv1 | 20 242 | 47 236 | $1.6 \times 10^{-3}$ | Logistic | Hessian | 132 | 127 | 137 |
| rcv1 | 20 242 | 47 236 | $1.6 \times 10^{-3}$ | Logistic | Working | 266 | 258 | 275 |
| scheetz | 120 | 18 975 | 1 | Least-Squares | Blitz | 0.706 | 0.689 | 0.722 |
| scheetz | 120 | 18 975 | 1 | Least-Squares | Celer | 0.801 | 0.777 | 0.826 |
| scheetz | 120 | 18 975 | 1 | Least-Squares | Hessian | 0.369 | 0.354 | 0.383 |
| scheetz | 120 | 18 975 | 1 | Least-Squares | Working | 0.643 | 0.639 | 0.647 |
| YearPredictionMSD | 463 715 | 90 | 1 | Least-Squares | Blitz | 706 | 704 | 707 |
| YearPredictionMSD | 463 715 | 90 | 1 | Least-Squares | Celer | 712 | 711 | 714 |
| YearPredictionMSD | 463 715 | 90 | 1 | Least-Squares | Hessian | 78.8 | 78.1 | 79.5 |
| YearPredictionMSD | 463 715 | 90 | 1 | Least-Squares | Working | 541 | 516 | 565 |

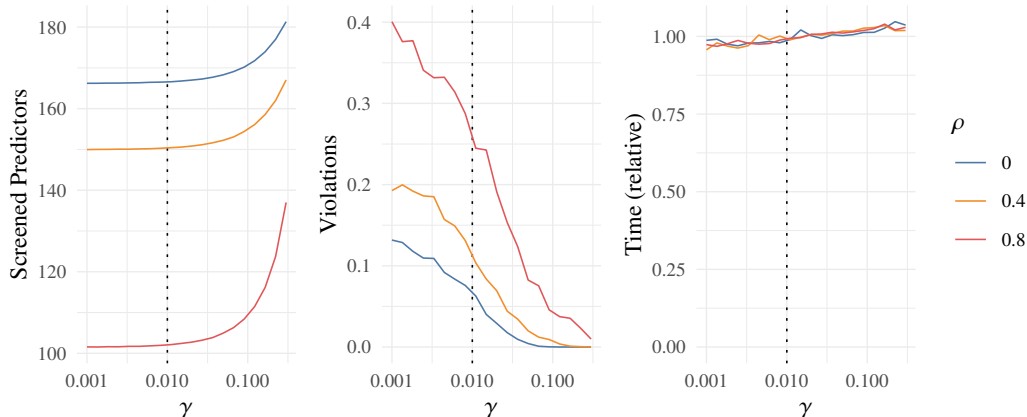

Figure 6: The number of predictors screened (included), the number of violations, and the time taken to fit the full path. All measures in the plots represent means across combinations of $\rho$ and $\gamma$ over 50 iterations. The time recorded here is the time relative to the mean time for each level of $\rho$. The choice of $\gamma$ in this work, 0.01, is indicated by a dotted line in the plots. Note that $x$ is on a $\log_{10}$ scale.

2. Hessian warm starts

3. Effective updates of the Hessian matrix and its inverse using the sweep operator

4. Gap safe screening

We then run an experiment on a design with $n = 200$ and $p = 20\,000$ and two levels of pairwise correlation between the predictors. The results (Figure 7) show that both screening and warm starts make considerable contributions in this example.

Note that these results are conditional on the order with which they are added and also on the specific design. The Hessian updates, for instance, make a larger contribution when $\min\{n, p\}$ is larger and $n$ and $p$ are more similar. And when $n \gg p$, the contribution of the warm starts dominate whereas screening no longer plays as much of a role.

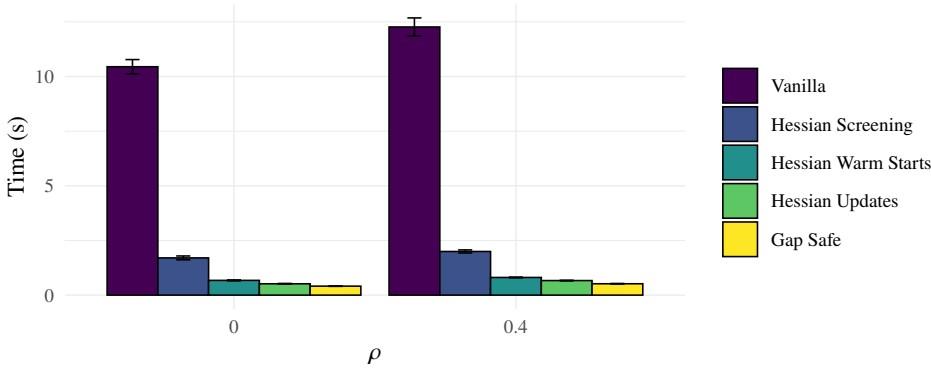

Figure 7: Incremental contribution to the decrease in running time from Hessian screening, Hessian warm starts, our effective updates of the Hessian and its inverse, and gap safe screening. In other words, *Gap Safe*, for instance, includes *all* of the other features, whilst *Hessian Warm Starts* includes only *Hessian Screening*. *Vanilla* does not include any screening and only uses standard warm starts (from the solution at the previous step along the path). The example shows an example of ordinary (least-squares) lasso fit to a design with $n = 200$ and $p = 20\,000$ with pairwise correlation between predictors given by $\rho$. (See Section 4 for more details on the setup). The error bars indicate standard 95% confidence intervals. The results are based on 10 iterations for each condition.

### F.9 $\ell_1$-Regularized Poisson Regression

In this experiment, we provide preliminary results for $\ell_1$-regularized Poisson regression. The setup is the same as Section 4 except for the following remarks:

- The response, $y$, is randomly sampled such that $y_i \sim \text{Poisson}\left(\exp(x_i^T\beta)\right)$.

- We set $\zeta$ in the convergence criterion to $n + \sum_{i=1}^{n} \log(y_i!)$.

- We do not use the line search procedure from Blitz.

- Due to convergence issues for higher values of $\rho$, we use values $0.0$, $0.15$, and $0.3$ here. Tackling higher values of $\rho$ would likely need considerable modifications to the coordinate descent solver we use.

- The gradient of the negative Poisson log-likelihood is not Lipschitz continuous, which means that Gap safe screening [15] no longer works. As a result, we have excluded the Blitz and Celer algorithms, which rely on Gap safe screening, from these benchmarks, and deactivated the additional Gap safe screening from our algorithm.

The results from the comparison are shown in Figure 8, showing that our algorithm is noticeably faster than the working algorithm also in this case.

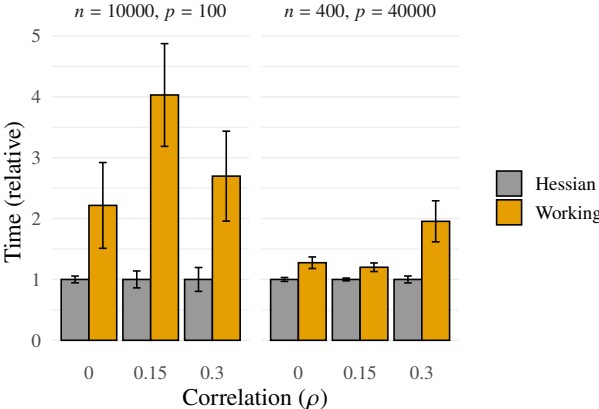

Figure 8: Time to fit a full regularization path for $\ell_1$-regularized Poisson regression to a design with observations, $p$ predictors, and pairwise correlation between predictors of $\rho$. Time is relative to the minimal mean time in each group. The error bars represent ordinary 95% confidence intervals around the mean.

### F.10 Runtime Breakdown Along Path

In this section we take a closer look at the running time of fitting the full regularization path and study the impact the Hessian screening rule and its warm starts have on the time spent on optimization of the problem using coordinate descent (CD).

To illustrate these cases we take a look at three data sets here: *e2006-tfidf*, *madelon*, and *rcv1*. The first of these, e2006-tfidf, is a sparse data set of dimensions $16\,087 \times 150\,360$ with a numeric response, to which we fit the ordinary lasso. The second two are both data sets with a binary response, for which we use $\ell_1$-regularized logistic regression. The dimensions of madelon are $2000 \times 500$ and the dimensions of rcv1 are $20\,242 \times 47\,236$.

We study the contribution to the total running time per step, comparing the Hessian screening rule with the working+ strategy. For the working+ strategy, all time is spent inside the CD optimizer and in checks of the KKT conditions. For the Hessian screening rule, time is also spent updating the Hessian and computing the correlation estimate $\tilde{c}^H$.

Beginning with Figure 9 we see that the Hessian strategy dominates the Working+ strategy, which spends most of its running time on coordinate descent iterations, which the Hessian strategy ensures are completed in much less time.

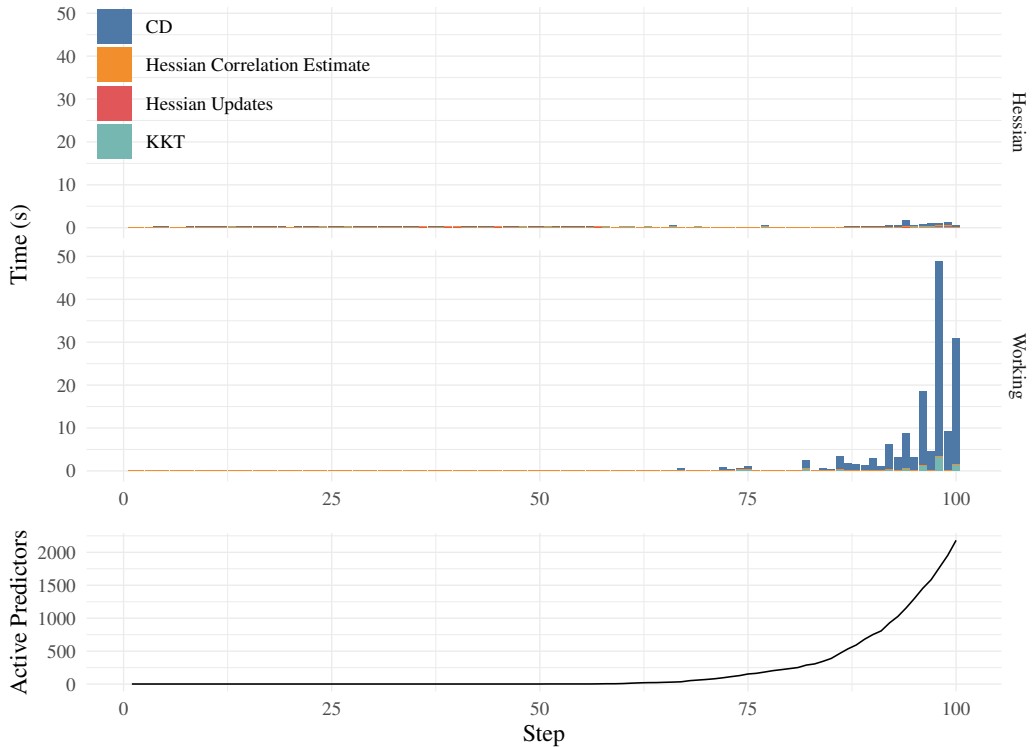

Figure 9: Relative contribution to the full running time when fitting a complete regularization path to the *e2006-tfidf* data set.

In Figure 10, we see an example of $\ell_1$-regularized logistic regression. In this case updating the Hessian exactly (and directly) dominates the other approaches. The size of the problem makes the cost of updating the Hessian negligible and offers improved screening and warm starts, which in turn greatly reduces the time spent on coordinate descent iterations and consequently the full time spent fitting the path.

Finally, in Figure 11 we consider the *rcv1* data set. In contrast to the case for *madelon*, the cost of directly forming the Hessian (and inverse) proves more time-consuming here (although the benefits still show in the time spent on coordinate descent iterations).

As a final remark, note that the pattern by which predictors enter the model (bottom panels) differ considerably between these three cases (Figures 9 to 11). Consider, for instance, *madelon* viz-a-viz *e2006-tfidf*. In *Approximate Homotopy* (Section 3.3.6), we discuss a remedy for this solution that is readily available through our method.