# OpenReview forum: "The Hessian Screening Rule"
_NeurIPS.cc/2022/Conference — NeurIPS 2022 Accept_

### Official Review · Reviewer_cyeA · 2022-07-09

**Rating:** 3
**Confidence:** 3
**Soundness:** 2 fair
**Presentation:** 3 good
**Contribution:** 1 poor

**Summary:**

The paper proposes a screening strategy for L1 sparse modelings. The basic idea is to use a prediction of an optimal solution as a function of the regularization parameter, derived from optimal conditions (which is aka solution path). The authors further propose combining working set selection screening by predicted solution with so-called strong rule to make screening more efficient.

**Questions:**

Convergence to the optimal solution is not mentioned. Could you give any information about the optimality of solution in a sense of the KKT conditions of (1)?

**Limitations:**

In Section 5, limitations were discussed.

**Strengths And Weaknesses:**

Overall, the paper is easy to follow and technical quality is fine. The purpose is clear and the procedure is written in detail. However, a critical issue is that the idea of main proposal (Hessian screening) is not novel. Approaches based on a similar idea have been studies in the context of the path following though it is not fully mentioned. Detailed comments are as follows.

Closely related papers in the path following literature are missed. For example, since the following two papers contains conceptually quite similar approaches, the difference should have been discussed in detail, though currently nothing is mentioned:
[Rosset2004] S. Rosset, Following Curved Regularized Optimization Solution Paths, NeurIPS 2004.
[Park2006] M. Y. Park and T. Hastie, L1 Regularization Path Algorithm for Generalized Linear Models, Journal of the Royal Statistical Society: Series B (Statistical Methodology), 2006.

The proposed algorithm can be seen as a so-called 'predictor-corrector' method in general (e.g., discussed in [Park2006]). Further, [Park2006] also discusses the working set selection based on the update equation (6).

Therefore, most importantly, the idea using Theorem 3.1 to predict the variable c (working set criterion) of the next \lambda has been already known. Therefore, I do not think the concept of 'Hessian screening' is novel.

[Rosset2004] also discusses a similar idea of approach (Hessian based update) based on an essentially quite similar theoretical property to Theorem 3.1. Further, this paper also provides the error analysis of the predictor.

Combination with strong rule and additional adjustment would be novel, but its technical significance is a bit marginal because these are quite simple heuristics.

Techniques in the 'Updating the Hessian' paragraph have been also known (the same techniques repeatedly discussed in the path following literature).

The 'Warm Starts' has also been widely known (e.g., [Park2006]).

Minor comments:
- Since Theorem 3.1 has been widely known, it should clarify more explicitly rather than noting it only in the footnote.

---

> ### Author Response · Authors · 2022-08-02
> **Response to cyeA**
>
> Thank you for your valuable comments and feedback!
>
> We agree that it was an oversight on our part not to cite Park and Hastie (2006) since we use the same type of warm starts, and possibly also Rosset (2004) given the paper’s theoretical discussion. We will amend this in the revision of our paper. Note, however, that it is not true that Park and Hastie (2006) use or even discuss working set strategies and/or screening rules. In step 3 of Lemma 1, the authors clearly state that the exact solution of $\beta$ is retrieved. It would moreover be surprising (at least to us) if working set or screening methods were used in this paper, given that Hastie is a co-author of the Strong Rule paper [10] that emerged several years later and proposes a screening strategy that our Hessian-based method clearly outperforms (see for instance figures 1 and 4). Nor does the Strong Rule paper mention Park and Hastie (2006) in any way. The paper does, however, discuss the poor performance of the strong rule for correlated predictors, which is exactly one of the issues that we address with our method.
>
> Both of the methods from these papers also lack many of the algorithmic strategies that we use in order to make the method feasible. In addition, please note that Rosset’s method is only applicable to $n > p$ and has a computational complexity (see section 2.2) that makes it computationally infeasible even in the case when $n > p$. As an example of this, we tried disabling our screening and effective Hessian updates when fitting the lasso path to the e2006-tfidf data set. With our method, using screening and effective Hessian updates, fitting the full path takes 12.6 seconds. Without these features, using only the methods from Park and Hastie (Hessian warm starts), the same path takes 129.9 seconds to fit. Please also see the additional experiments that we have now added to the supplement, in particular the ablation analysis in F.7.
>
> > Convergence to the optimal solution is not mentioned. Could you give any information about the optimality of solution in a sense of the KKT conditions of (1)?
>
> Could you please clarify what you mean here? In Section 4, paragraph 4, we state that we consider convergence in terms of the duality gap, which explicitly provides the difference between the optimal and current solution. The KKT conditions and duality gap are intimately related, but we are happy to make this connection more clear in the final revision.

---

> > ### Comment · Reviewer_cyeA · 2022-08-08
> > **thank you for you reply**
> >
> > Thank you for your reply.
> >
> > > Note, however, that it is not true that Park and Hastie (2006) use or even discuss working set strategies and/or screening rules.
> >
> > [Park2006] discusses the working set selection (though they call it 'active set', which is often used interchangeably). In each step of the regularization path, the solution is predicted by the essentially same approximation, and after correcting the predicted solution, the working set (active set) is updated based on the KKT condition (In sec2.2.3, 'repeat the corrector step with the modified active set until the active set is not augmented further'. This indicates corrector step optimizes variables in the active set and update the set if the optimality is not satisfied). Note that I did not say [Park2006] discussed strong rule.
> >
> > > KKT condition
> >
> > Section 4 (paragraph 4) is in the experiment section, and it does not clearly state if the proposed method is theoretically guaranteed. The convergence should have been discussed in the methodology section whether the final optimality (either KKT or duality gap) is theoretically guaranteed in the proposed method (perhaps, as a theorem).

---

> > > ### Author Response · Authors · 2022-08-09
> > > **Second response to cyeA**
> > >
> > > > [Park2006] discusses the working set selection (though they call it 'active set', which is often used interchangeably). In each step of the regularization path, the solution is predicted by the essentially same approximation, and after correcting the predicted solution, the working set (active set) is updated based on the KKT condition (In sec2.2.3, 'repeat the corrector step with the modified active set until the active set is not augmented further'. This indicates corrector step optimizes variables in the active set and update the set if the optimality is not satisfied).
> > >
> > > We have now looked through [the code for the Park and Hastie paper](https://github.com/cran/glmpath) and concede that the authors in fact do appear to have used a form of working strategy. Specifically, they use the same kind of working set that is used in the Strong Rule paper by Tibshirani et al, i.e. they use the active set from the previous step on the path as a working set. **This is not the same working set that we use.** Our working set is based on screening by estimating the correlation vector for the next step via second-order information. This procedure is not used in Park and Hastie's paper.
> > >
> > > We will naturally amend our paper accordingly to make proper reference to the use of working sets in Hastie and Park.
> > >
> > > Regarding the nomenclature, it is true that some authors use "active set" to mean "working set", but this is not the case with Park and Hastie. Note, for instance, that they on page 667 write that they propose strategies for "estimating where the active set changes" and on page 670 state that "the active set cannot exceed the sample size at any segment of the paths".
> > >
> > > > Note that I did not say [Park2006] discussed strong rule.
> > >
> > > We did not mean to suggest as much. We brought this up in order to emphasize that **our work is about screening rules** and that it would be highly unlikely for Hastie to have discovered a Hessian-based screening strategy years before authoring the Strong Rule paper yet make no reference to it.
> > >
> > > > Section 4 (paragraph 4) is in the experiment section, and it does not clearly state if the proposed method is theoretically guaranteed. The convergence should have been discussed in the methodology section whether the final optimality (either KKT or duality gap) is theoretically guaranteed in the proposed method (perhaps, as a theorem).
> > >
> > > Thank you for the clarification. We believed this point to be clear from our paper given the strategy we use for updating our working sets: adding predictors that fail the KKT checks into the working set and refitting. But we now see that this isn't the case and will happily make this more clear through a theorem or proposition in the final revision.

---

> ### Author Response · Authors · 2022-08-08
> **Further questions**
>
> Please let us know if there are still any outstanding issues that you would like to discuss or if anything in our last response was unclear.

---

### Official Review · Reviewer_ERSS · 2022-07-10

**Rating:** 6
**Confidence:** 3
**Soundness:** 2 fair
**Presentation:** 3 good
**Contribution:** 2 fair

**Summary:**

This paper studies the predictor screening rules over the lasso optimization problem. It proposes a Hessian screening rule which utilizes the second-order information. This rule is effective not only in screening but also in accurate warm starts. Updating the second-order information has high computational complexity, and to deal with it, this work replies on the sweep operator. In the experiments, the proposed rule is compared to many baselines and outperforms them significantly on both simulated and real-world data.



**Questions:**

See the concerns in the above section.

**Limitations:**

Yes

**Strengths And Weaknesses:**

Originality

Previous works on screening rules overlooked the study on the second-order information. Thus I think the direction of this work novel. The proposed rule takes the warm start and the Hessian matrix computation problems into account, and resolves them soundly. Besides, the insightful discussions on the proposed screening rule are provided. They are helpful to understand the rule and to differentiate this approach with previous methods. Therefore, I think the contributions of the work novel.

Quality

I am satisfactory with most contents of this paper. The paper presents a clear overview of the question and previous rules both in words and math. The proposed rule is based on the Hessian matrix and is actually in a form of the second-order Taylor approximation. Speeding up the Hessian matrix computation is based on the sweep operator and the warm starts also benefit from the Hessian matrix. These arguments are demonstrated in the experiments. As far as I checked, the theoretical analyses have no problem.

However, I have a concern with the study method. Normally, people select the predictors for better fitting accuracy, but this paper relies heavily on the time cost and the minimum number of active predictors to measure the performance. I am curious about why not including fitting accuracy.

Clarity

The paper is clearly written for me. I can easily follow the contents on the approach and the experiment.

Significance

The idea of the approach is sound and practical performance is better than baselines. However, because just the lasso problem is involved, I am afraid that the audience will be very limited. Therefore, I think the significance of this paper is limited.

---

> ### Author Response · Authors · 2022-08-02
> **Response to ERSS**
>
> Thank you for your encouraging and insightful comments!
>
> > However, I have a concern with the study method. Normally, people select the predictors for better fitting accuracy, but this paper relies heavily on the time cost and the minimum number of active predictors to measure the performance. I am curious about why not including fitting accuracy.
>
> We are not certain that we understand fully what you mean here, so please elaborate if that is the case. The screening strategy is used only for improving the speed and does not change the accuracy of fitting the lasso.
>
> > However, because just the lasso problem is involved, I am afraid that the audience will be very limited. Therefore, I think the significance of this paper is limited.
>
> We do not quite agree that the relevance of the paper is limited. It is true that our submission concerns screening specifically for the lasso. But our method, using second order information (the Hessian), can in principle be used for many other objectives as well. Extensions to the elastic net are for instance straightforward and we also (in principle) know how to extend it to Sorted L-One Penalized Estimation (SLOPE/OWL) too (including OSCAR). Extensions to MCP and SCAD may also be possible.

---

> > ### Comment · Reviewer_ERSS · 2022-08-08
> > **Response to Authors**
> >
> > Thanks for the authors' replies. After I read other reviews and replies, I realize that my concern about fitting accuracy is similar to that of Reviewer ovwH. I wonder about the fitting accuracy when using different screening rules, especially on real-life data. If the Hessian rule has the advantages of efficiency and high accuracy, it will be wonderful.

---

> > > ### Author Response · Authors · 2022-08-08
> > > **Second response to ERSS**
> > >
> > > Please note that this is a misconception. There is no difference in the accuracy between the heuristic methods (working, strong, and Hessian) and the so-called SAFE methods. We always check the KKT conditions upon convergence on the working set obtained by these methods and add each predictor that fail this test to the working (screened) set. Only when the duality gap for the full set of predictors reaches our tolerance level do we consider the solver to converge and we use this criterion for all of the methods in our experiments. This means that the theoretical guarantee that we end up with the actual active set is equally strong for our method.

---

> > > ### Author Response · Authors · 2022-08-09
> > > **Additional comment regarding convergence**
> > >
> > > Since several reviewers have now brought up discussions regarding theoretical guarantees for convergence, we recognize that this could be clarified in the paper. We will therefore add a theorem or proposition on the convergence of our method to the final revision, showing that our method converges towards the true solution.

---

### Official Review · Reviewer_ovwH · 2022-07-10

**Rating:** 6
**Confidence:** 3
**Soundness:** 3 good
**Presentation:** 3 good
**Contribution:** 3 good

**Summary:**

A Hessian screening rule for lasso and its generalized linear model extension for logistic regression was presented to take advantage of the high-order information for more efficient screening, specifically in cases with highly correlated predictors/covariates. The proposed Hessian screening rule together with several speedup tricks has been shown to be effective in both simulated and real datasets.

**Questions:**

1. Are there any theoretical guarantee that the Hessian rule will be always less conservative than the Strong rule or Working Set method?

2. How the corresponding prediction accuracy of the resulting models by different screening rules? Are the final "screened" predictors all the same as the ones from the optimal solution from the model fitting without screening?

3. There are multiple screening rules published more recently, including: https://proceedings.neurips.cc/paper/2020/hash/11348e03e23b137d55d94464250a67a2-Abstract.html, how does the Hessian screening rule compare with these new rules?

**Limitations:**

N/A.

**Strengths And Weaknesses:**

The proposed Hessian screening rule extends the Strong Rule and Working Set by taking advantage of the high-order information for more efficient screening, specifically in cases with highly correlated predictors/covariates. The Hessian screening rule together with several speedup tricks has been shown to be effective in both simulated and real datasets.

It is not clear based on the current presentation why the Hessian rule can be less conservation from Section 3. In addition to Theorem 3.1, some theoretical analysis for that may help further improve the quality of the submission. The actual final screening in fact was based on the modifications as described from line 133 to 151. It may be interesting also to have ablation comparison to see clearly what led to improved efficiency.

Since, the screening rules are not "safe", in particular for logistic regression. In addition to investigating the efficiency, the authors may also need to provide the performance comparison with respect to both predictor selection and model prediction accuracy.

In the real data experiments, the authors may want to provide some explanations why the Hessian rule performs significantly worse on arcene and rcv2 datasets, for which p is much larger than n, especially for arcene. It is clearly not the case when p is similar as n as discussed in Section 5.

Finally, there are language problems in the submission. For example, in line 188-189 (page 5): "... this is not a surprising find."  In line 257-258 (page 7), "we also stop the path whenever the number of coefficients ever to be active predictors exceeds p." The number of coefficients can be equal to p but will never exceed p. The authors may need to improve the presentation of the submission.

---

> ### Author Response · Authors · 2022-08-02
> **Response to ovwH**
>
> Thank you for your detailed comments and relevant questions!
>
> >  It may be interesting also to have ablation comparison to see clearly what led to improved efficiency.
>
> We have added two additional experiments to the supplement that tackle this. Please see the general comment for more information.
>
> > In the real data experiments, the authors may want to provide some explanations why the Hessian rule performs significantly worse on arcene and rcv2 datasets
>
> Please see the general comment and the updated results, in which our method now outperforms the other methods on the rcv1 data set as well and performs on par with the other methods for arcene.
>
> > There are multiple screening rules published more recently [...] how does the Hessian screening rule compare with these new rules?
>
> Thank you for drawing our attention to this method. We have compared our method with the author’s code for Thunder for fitting the full path for two of the data sets from our paper.
>
> ```
> Data Set    Hessian Thunder
> ----------- ------- -------
> rcv1          28.20   37.85
> news20       162.94  313.82
> ```
> The Hessian screening rule outperforms Thunder in every instance. We also attempted to use the Thunder code for the data sets arcene, e2006-tfidf, and e2006-log1p, but encountered convergence issues.
>
> Note, however, that these experiments cannot be compared with the ones in our paper since we have had to
>
> 1. change standardization from centering by the mean and scaling by standard deviation to only scaling by the maximum absolute value of each column,
> 2. use only the ordinary (least-squares) lasso, and
> 3. restrict these initial experiments to sparse designs
>
> because the Thunder implementation only supports this setup. Note also that there are differences in the implementations. Both are based on C++ but use different linear algebra routines as well as implementations of the coordinate descent solver (we use line search but Thunder does not). We also note that there are some outstanding issues with the code that we would need to fix if we were to include Thunder in our final revision. We for instance encountered cases where the $R_1$ set in the algorithm would become empty, causing the algorithm to loop indefinitely without converging.
>
> We are somewhat surprised by the results from the Thunder paper, in particular the comparisons with Celer given that the two methods are so similar. Both Celer and Thunder use gap safe screening, working sets, extrapolated duals, and pruning (even though this feature of Celer is not used in the Thunder benchmarks). The distinguishing feature as we understand it is that Thunder uses more advanced algorithmic techniques to construct its working sets. Using our implementation of Celer, we see similar results compared to Thunder. One possible reason for this may be that the Thunder code compiles its data sets into binary code before fitting, which should lead to better performance, yet does not include the time required for this compilation into its benchmarks.
>
> You also mention that there are “multiple” screening rules published recently. If you think there are other recent screening rules that we might have missed, then we would be grateful if you would share this with us. We acknowledge that it is unfortunate that we overlooked Thunder but have upon conducting new searches not been able to find any other relevant screening strategy.
>
> > Are there any theoretical guarantee that the Hessian rule will be always less conservative than the Strong rule or Working Set method?
>
> We believe that there is not, but it is an interesting question that should be explored further. If space allows, we will add some analysis of this aspect in our final revision.
>
> > How the corresponding prediction accuracy of the resulting models by different screening rules? Are the final "screened" predictors all the same as the ones from the optimal solution from the model fitting without screening?
>
> We take it to mean that you mean “active” rather than “screened” predictors, since all the rules are safe in practice and always consider all of the predictors before converging. In this case the answer is no: the active set may differ between the different screening rules since the solver may converge on a solution with a global duality gap (taking all predictors into consideration) below the threshold. Note that this criterion is also used in Celer and Blitz for instance. The tolerance threshold we use in our experiments, however, ensures that the active sets of the screening rules are always very similar among the screening rules.
>
> > [T]here are language problems in the submission. For example, in line 188-189 (page 5): "... this is not a surprising find."
>
> We do not see the error here. Could you please clarify? Note that “find” is a noun in this sentence.
>
> > The number of coefficients can be equal to p but will never exceed p.
>
> Thanks for highlighting this typo. We will revise the paper accordingly.

---

> > ### Comment · Reviewer_ovwH · 2022-08-08
> > **screening performance**
> >
> > I thank the authors' additional experiments to check the runtime and compare to more recent algorithms. As there are quite many screening rules for lasso in the literature, it would be definitely good to have more comprehensive performance comparison results by considering existing rules, for example, the ones reviewed and introduced in: https://dl.acm.org/doi/10.1109/TPAMI.2017.2765321.
> >
> > More critically, as there is no theoretical guarantee that the proposed Hessian screening rule will end up with the actual "active" predictors, it is important to compare the accuracy of the resulting models in addition to runtime comparison. It will be not really meaningful to have a fast screening method but end up with the models that do not perform well.

---

> > > ### Author Response · Authors · 2022-08-08
> > > **Second response to ovwH**
> > >
> > > > As there are quite many screening rules for lasso in the literature, it would be definitely good to have more comprehensive performance comparison results by considering existing rules, for example, the ones reviewed and introduced in: https://dl.acm.org/doi/10.1109/TPAMI.2017.2765321.
> > >
> > > Thank you for this additional reference. If there are any specific rules that you would want us to include in our comparisons from this article then we would be happy to do so. Note, however, that we are already included many of the rules from this paper or, alternatively, rules which supersede them. The SAFE, sequential SAFE, DOME, Sphere, and ST3 tests, for instance, have been shown to be inferior to Gap-Safe screening rules, which we have included. (See Ndiaye et al. [7] for several experiments that demonstrate this.) Sasvi screening, which is also mentioned in the article, has been superseded by dynamic Sasvi screening [8], which we also considered for our submission (see for instance figure 1). We have also included the Strong rule and EDPP (see figure 1). Observe that EDPP directly supersede DPP. Finally, note also that EDPP rules are in fact not safe in practice since they rely on the exact solution from the previous step on the path, which is never available in practice.
> > >
> > > > More critically, as there is no theoretical guarantee that the proposed Hessian screening rule will end up with the actual "active" predictors, it is important to compare the accuracy of the resulting models in addition to runtime comparison. It will be not really meaningful to have a fast screening method but end up with the models that do not perform well.
> > >
> > > Please note that this is a misconception. There is **no difference** in the accuracy between the heuristic methods (working, strong, and Hessian) and the so-called SAFE methods. We always check the KKT conditions upon convergence on the working set obtained by these methods and add each predictor that fail this test to the working (screened) set. Only when the duality gap for the full set of predictors reaches our tolerance level do we consider the solver to converge and we use this criterion for all of the methods in our experiments. This means that **the theoretical guarantee that we end up with the actual active set is equally strong for our method**.

---

> > > ### Author Response · Authors · 2022-08-09
> > > **Additional comment regarding theoretical guarantees for convergence**
> > >
> > > Since several reviewers have now brought up discussions regarding theoretical guarantees for convergence, we recognize that this could be clarified in the paper. We will therefore add a theorem or proposition on the convergence of our method to the final revision, showing that our method converges towards the true solution.

---

### Official Review · Reviewer_B3d7 · 2022-07-10

**Rating:** 6
**Confidence:** 3
**Soundness:** 3 good
**Presentation:** 3 good
**Contribution:** 2 fair

**Summary:**

The paper proposes a heuristic (non-safe) screening rule to deal with l1-regularized estimation problems such as linear regression and logistic regression. The proposed method can be viewed as a generalization of the strong rule used in the glmnet, which only used first-order information.  The paper reports experiments on both synthetic data and real-world data to illustrate the performance of the proposed method.

**Questions:**

I don't have additional questions for the authors.

**Limitations:**

Limitations are discussed in the paper.

**Strengths And Weaknesses:**

Strengths:
* the paper addresses an interesting problem in l1-regularized estimation for linear regression and logistic regression. Screening rules are an effective strategy to speed up such estimation.
* Although the proposed rule is heuristic in nature, the simplicity in its formulation and the effectiveness shown in the experiments offers some advantages of the proposed method. This is also a not-so-common approach that makes use of second-order information for screening.
* Experiments are exhaustive. six well-known alternative methods are compared on a wide variety of synthetic and real-world data.

Weakness:
* The authors may consider carrying out experiments on other settings covered by the proposed method such as poison regression and elastic net. It may also be interesting (somewhat orthgonal) to understand how the proposed method performs or scale compared to SGD based approach on very large datasets.

---

> ### Author Response · Authors · 2022-08-02
> **Response to B3d7**
>
> Thank you for your encouraging comments and important questions!
>
> > The authors may consider carrying out experiments on other settings covered by the proposed method such as poison regression and elastic net.
>
> We agree that it would be interesting to also study Poisson regression. Due to the limited amount of time for the rebuttal, however, we will not be able to add results on this at this time but will gladly do so in time for the final revision of the paper.
>
> We also agree that results for elastic net would be of some interest but will not have time to add this to our paper, although we will be happy to add a paragraph to the paper discussing extensions to the elastic net. Note, however, that the addition of a ridge penalty would likely only benefit our method by making the overall cost function more quadratic, improving the Hessian estimate both in terms of accuracy and stability. With the addition of the ridge penalty, the Hessian matrix changes from $X_A^T X_A$ to $X_A^T X_A + \lambda I$, where $\lambda$ is the ridge penalty. This will increase the stability of the matrix in the sense that the condition number of $X_A^T X_A + \lambda I$ is smaller than  $X_A^T X_A$. How to incorporate an efficient update of the Hessian is not, however, as clear, although at first glance it seems that it should be possible.
>
> > It may also be interesting (somewhat orthgonal) to understand how the proposed method performs or scale compared to SGD based approach on very large datasets.
>
> That is a question that would be very interesting to examine further but which we think may be out of scope for this work. For instance, we are aware of papers where mini-batches are selected to guarantee positive definiteness of the Hessian and it would be interesting to combine this method with our algorithm. We will add a sentence regarding this to the discussion in the final revision.

---

### Author Response · Authors · 2022-08-02
**General Response to All Reviewers**

We would first like to thank all of the reviewers for their hard work and constructive feedback!

We recently discovered that our implementation of the Hessian updates involved the accidental construction of a dense diagonal matrix, which caused our method to underperform for the experiments with sparse designs and $\ell_1$-regularized logistic regression. Most importantly, **note that this only affected our method**. We have now re-run the experiments on real data and note that our method now performs best also for rcv1, which was previously a notable exception in our results. Please see the paper for updated results. Note that the updated results differ slightly from the ones in the submitted version due to changes on the benchmarking machine, but that these changes do not alter any of our conclusions. We have also updated the results section slightly as well as the second paragraph of the discussion in light of these changes.

In addition to this, we have also extended our results in the supplementary with two more experiments:

- In F.7 (supplement) we provide an ablation analysis which investigates how the various features of our screening algorithm influence the time to fit a regularization path. For the final revision we will extend this experiment to additional scenarios.
- In F.8 (supplement) we provide a runtime breakdown for the entire path to study how time is spent along the path on screening, Hessian updates, and optimization (coordinate descent) for three data sets.
We believe that these two experiments provide additional insight into the performance of our method and hope that the reviewers are able to spare the time to consider them.

---

### Meta-Review · Area_Chair_DvHB · 2022-08-27

**Recommendation:** Accept
**Confidence:** Less certain

**Metareview:**

This papers proposed a Hessian screening rule for lasso and its generalized linear model extension for logistic regression. The proposed screening rules have been demonstrated to be effective in both simulated and real datasets. The idea is novel and the evaluation is convincing. The authors mention that extensions to MCP and SCAD may also be possible, even though the objective may not be convex. A brief discussion will be helpful.

**Award:**

No

---

### Decision · Program_Chairs · 2022-09-14

Accept